# WATS: Wavelet-Aware Temperature Scaling for Reliable Graph Neural Networks

**Xiaoyang Li**
Independent Researcher
Lxy289692485@gmail.com

**Linwei Tao**
University of Sydney
linwei.tao@sydney.edu.au

**Haohui Lu**
Charles Darwin University
haohui.lu@cdu.edu.au

**Minjing Dong**
City University of Hong Kong
minjdong@cityu.edu.hk

**Junbin Gao**
University of Sydney
junbin.gao@sydney.edu.au

**Chang Xu**
University of Sydney
c.xu@sydney.edu.au

## Abstract

Graph Neural Networks (GNNs) have demonstrated strong predictive performance on relational data; however, their confidence estimates often misalign with actual predictive correctness, posing significant limitations for deployment in safety-critical settings. While existing graph-aware calibration methods seek to mitigate this limitation, they primarily depend on coarse one-hop statistics, such as neighbor-predicted confidence, or latent node embeddings, thereby neglecting the fine-grained structural heterogeneity inherent in graph topology. In this work, we propose Wavelet-Aware Temperature Scaling (WATS), a post-hoc calibration framework for node classification that assigns node-specific temperatures based on tunable heat-kernel graph wavelet features. Specifically, WATS harnesses the scalability and topology sensitivity of graph wavelets to refine confidence estimates, all without necessitating model retraining or access to neighboring logits or predictions. Extensive evaluations across nine benchmark datasets with varying graph structures and three GNN backbones demonstrate that WATS achieves the lowest Expected Calibration Error (ECE) among most of the compared methods, outperforming both classical and graph-specific baselines by up to 41.2% in ECE and reducing calibration variance by 15.84% on average compared with graph-specific methods. Moreover, WATS remains computationally efficient, scaling well across graphs of diverse sizes and densities. The implementation is available at https://github.com/lxy1134/WATS.git

## 1 Introduction

GNNs offer a principled approach for learning over structured data and have achieved strong empirical performance across a wide range of domains, including social network modeling (Fan et al., 2019), traffic forecasting (Sharma et al., 2023), and healthcare applications (Gao et al., 2024; Lu & Uddin, 2021). They support key tasks such as node classification (Zhao et al., 2021; Sun et al., 2022), link prediction (Zhang & Chen, 2018; Luo et al., 2023), and graph-level inference (Zhang et al., 2021; Godwin et al., 2021). While GNNs are widely adopted for their representational power, their output confidence often fails to reflect true predictive reliability, which is an issue of growing concern in high-stakes domains, for instance, medical diagnosis and financial risk assessment.

Model calibration, which measures the alignment between a model's predicted confidence and its true correctness likelihood (Guo et al., 2017), is a key aspect of model reliability. A well-calibrated model is expected to produce predictions whose confidence scores accurately reflect the observed accuracy. For example, a prediction made with 70% confidence should be correct 70% of the time. Recent findings highlight that GNNs behave differently from standard independent and identically distributed ($i.i.d.$) trained models such as CNNs and transformers (Melotti et al., 2022; Tao et al.,

2025; Wen et al., 2024). Unlike CNNs and transformers, which often suffer from overconfidence, GNNs tend to be systematically underconfident: their predicted confidence scores are consistently lower than their true accuracy (Wang et al., 2022; Hsu et al., 2022; Liu et al., 2022).

Several recent approaches aim to address this issue through graph-aware calibration strategies, including Graph Attention Temperature Scaling (GATS) (Hsu et al., 2022), CaGCN (Wang et al., 2021), Graph Ensemble Temperature Scaling (GETS) (Zhuang et al., 2024), and SimCalib (Tang et al., 2024). These approaches typically enhance node-level calibration by integrating neighbour structural cues with their predictive status, such as confidence level.

However, they predominantly rely on shallow neighborhood statistics or opaque latent representations, which may result in unstable and inaccurate uncertainty estimation. As they capture only limited, local information and fail to reflect the broader structural context, leading to unreliable calibration, especially for low-degree scenarios. For illustration, we provide empirical evidence that nodes with similar local statistics can exhibit different miscalibration levels across graphs. Based on the above limitation, we aim to build a method that: (i) flexibly incorporates neighborhood information without relying on additional explicit pretraining status (ii) maintains high calibration performance across diverse graph domains, while remaining lightweight and post-hoc. (iii) performs temperature scaling at the node level to allow identical correction based on multi-hop structural information.

Therefore, in this work, we propose a calibration framework for node classification, called WAVELET-AWARE TEMPERATURE SCALING (WATS), which introduces flexibly scaled structural features through graph wavelets. We incorporate graph wavelets because they offer a principled way to capture structural information at multiple scales (Donnat et al., 2018). Our methods, wavelets spatial localization controlled by the scale parameters $s$ and $k$, enabling fine-grained capture of multi-hop dependencies. Differs from graph wavelet algorithms used in neural network (Donnat et al., 2018; Behmanesh et al., 2022) in that it does not aim to reconstruct or smooth node features for node classification. Instead, it uses wavelet coefficients as structural signatures to indicate the node uncertainty, which allows WATS to adaptively calibrate predictions based on the underlying structure, helping to correct confidence errors where standard methods fall short. Our main contributions are summarized as follows:

**WATS**: We propose a novel post-hoc calibration framework that performs node-wise temperature scaling using flexibly scaled graph wavelet features.

**Robust, interpretable structural features:** We show that graph wavelet features are stable and geometry-aware, encoding local graph structure without relying on potentially noisy signals such as neighboring logits or distances to labeled nodes.

**Extensive empirical validation:** Across multiple graph benchmarks and GNN architectures. WATS consistently improves calibration quality, reducing ECE and outperforming both classical and graph-specific calibration baselines.

## 2 RELATED WORK

### 2.1 UNCERTAINTY CALIBRATION FOR NEURAL NETWORKS

Uncertainty calibration aims to align a model's predicted confidence with the true likelihood of correctness. Existing approaches are typically divided into two categories: in-training and post-hoc

**In-training** approaches incorporate uncertainty estimation within the model optimization process. For example, Bayesian Neural Networks (BNNs) and variational inference methods achieve this by imposing probabilistic distribution over model parameters (Gal & Ghahramani, 2016; MacKay, 1995; Springenberg et al., 2016). Alternative frequentist strategies, such as quantile regression (Romano et al., 2019), are also employed to generate calibrated probability estimates.

**Post-hoc** methods, in contrast, calibrate a pre-trained model without modifying its internal parameters. These include non-parametric techniques such as histogram binning (Zadrozny & Elkan, 2001) and isotonic regression (Zadrozny & Elkan, 2002), parametric approaches that assume a specific transformation form, including temperature scaling (TS) (Guo et al., 2017) and Beta Calibration (Kull et al., 2017), as well as distribution-free frameworks like conformal prediction (Tibshirani et al., 2019) which provides valid coverage guarantees for model outputs.

## 2.2 GRAPH-SPECIFIC CALIBRATION METHODS

While post-hoc calibration methods perform well on Euclidean data with CNNs, their effectiveness declines on graph-structured data due to the lack of relational modeling (Hsu et al., 2022; Wang et al., 2021). To address this, several graph-aware approaches have been proposed. CaGCN (Wang et al., 2021) uses a GCN-based temperature predictor to incorporate structure, while GATS (Hsu et al., 2022) applies attention over neighborhoods for node-specific temperatures. GETS (Zhuang et al., 2024) introduces a sparse mixture-of-experts using degree, features, and confidence. SimCalib (Tang et al., 2024) adds similarity-preserving regularization, and Shi et al. (2023) use reinforcement learning to adapt calibration to graph structure. Beyond post-hoc methods, Yang et al. (2024a) reweight edges during training to improve calibration, and Yang et al. (2024b) propose a calibration-aware loss targeting underconfidence caused by shallow GNNs. These methods collectively integrate graph structure and node-level signals to enhance calibration.

## 2.3 GRAPH WAVELET

Graph wavelets provide compact, spatially localized bases that are well suited to graph signal processing and structural representation learning. Whereas classical wavelets such as Haar and Daubechies are defined on Euclidean lattices (Bruce et al., 1996; Resnikoff & Wells, 1992), graph wavelets extend these ideas to non-Euclidean domains by leveraging the spectral properties of the graph Laplacian (Das, 2004; Shuman et al., 2013; Xu et al., 2019).

A particularly versatile construction is the lifting scheme (Sweldens, 1998), which can be transferred to graphs without any data-driven training. Hammond et al. (2011) further improved practicality by replacing the costly Laplacian eigendecomposition with Chebyshev polynomial approximations, enabling efficient wavelet transforms on large graphs. Since then, graph wavelets have supported a variety of downstream tasks, including graph convolutional architectures (Xu et al., 2019; Deb et al., 2024), multimodal wavelet networks (Behmanesh et al., 2022), community detection via scale-adaptive filtering (Tremblay & Borgnat, 2014), and diffusion-based node embeddings that capture multi-scale structural patterns (Donnat et al., 2018). All these methods and application indicates the importance of Graph wavelet in both theoretical and empirically practice

# 3 METHOD

## 3.1 PRELIMINARY STUDY

We address the problem of uncertainty calibration in semi-supervised node classification tasks over graphs. Let $\mathcal{G} = (\mathcal{V}, \mathcal{E})$ denote a graph, where $\mathcal{V}$ is the set of nodes and $\mathcal{E}$ is the set of edges. The adjacency matrix is denoted by $A \in \mathbb{R}^{N \times N}$, where $N = |\mathcal{V}|$. Each node $v_i \in \mathcal{V}$ has a feature $x_i \in \mathcal{X}$, and for a subset of labeled nodes $\mathcal{L} \subseteq \mathcal{V}$, the true label $y_i \in \{1, \ldots, K\}$ is provided. Let $X = [x_1, \ldots, x_N]^\top$ be the feature matrix and $Y = [y_1, \ldots, y_N]^\top$ the label vector, a GNN $f_\theta$ performs node classification via predicting node-wise class probabilities $\hat{p}_i(y)$ as

$$\hat{y}_i = \arg \max_y \hat{p}_i(y), \quad \hat{c}_i = \max_y \hat{p}_i(y),$$

where $y_i$ denotes the predicted label and $c_i$ denotes its confidence. In the field of model calibration, a well-calibrated model provides confidence that aligns with the true accuracy well as

$$\mathbb{P}(y_i = \hat{y}_i \mid \hat{c}_i = c) = c \quad \forall c \in [0, 1].$$

The measurement of model calibration can be computed via Expected Calibration Error (ECE) (Guo et al., 2017) as $\mathbb{E}[|\mathbb{P}(y_i = \hat{y}_i|\hat{c}_i) - c|]$, however, ECE cannot be easily computed due to limited samples. Thus, an estimation of ECE is introduced by grouping samples into $M$ bins with equal confidence intervals as $B_m = \{ j \in \mathcal{N} \mid \frac{m-1}{M} < \hat{c}_j \leq \frac{m}{M} \}$, where $\mathcal{N}$ is the subset of node that used in evaluation $\mathcal{N} \subseteq \mathcal{V}$. Given the bin accuracy $\text{Acc}(B_m) = \frac{1}{|B_m|} \sum_{i \in B_m} \mathbf{1}(\hat{y}_i = y_i)$ and bin confidence $\text{Conf}(B_m) = \frac{1}{|B_m|} \sum_{i \in B_m} \hat{c}_i$, the approximation can be achieved by computing the expected difference between bin accuracy and confidence as

$$\text{ECE} = \sum_{m=1}^{M} \frac{|B_m|}{|\mathcal{N}|} \big| \text{Acc}(B_m) - \text{Conf}(B_m) \big|. \tag{1}$$

## 3.2 UNCERTAINTY ESTIMATION IN GNNS

**Calibrate via One-hop Statistics**   Message-passing is widely adopted in GNNS, including GCN (Kipf & Welling, 2016) and GAT (Veličković et al., 2017), which can be simplified via a degree-normalized mean aggregator as

$$h_i^{(\ell+1)} = \frac{1}{d_i + 1}\Big(h_i^{(\ell)} + \sum_{j \in \mathcal{N}(i)} h_j^{(\ell)}\Big), \quad d_i = |\mathcal{N}(i)|. \tag{2}$$

where $d_i$ excludes the node itself, so $d_i + 1$ accounts for the self-loop. The final embedding $h_i^{(L)}$ induces the confidence $\hat{c}_i$. This local aggregation implicitly determines the final prediction and the associated confidence $\hat{c}_i$ of node $i$. Although GATS confines all structural operations, including neighbor temperature aggregation, attention weights, neighbor confidence averaging to 1-hop, and CaGCN and GETS stack two GCN layers to nominally reach 2-hop, each layer itself still performs only 1-hop aggregation. As a result, these methods are unable to adaptively capture longer-range dependencies. Although these calibration techniques show effectiveness, we argue that one-hop statistics only cannot provide an accurate estimation of node uncertainty in GNNs. Considering a simplified one-hop estimator of confidence:

$$\hat{c}_i \approx \frac{1}{d_i + 1} \sum_{j \in \{i\} \cup \mathcal{N}(i)} y_j,$$

where $y_j \in \{0, 1\}$ is the true label indicator for node $j$. Then the per-node calibration bias is

$$\text{bias}_i = \big|\hat{c}_i - \mathbf{1}(\hat{y}_i = y_i)\big| \approx \Big| y_i - \frac{1}{d_i + 1} \sum_{j \in \mathcal{N}(i)} y_j \Big|. \tag{3}$$

As shown in Eq. 3, for example, when $d_i = 2$ and the neighbor labels are $[0, 1]$, the average is $1/3$ regardless of the true label $y_i$, making the estimate uninformative, which means high uncertainty. In sparse or low-homophily regions, one-hop neighborhoods may carry weak or misleading signals, resulting in poor bias approximation. This motivates structure-aware calibration that aggregates richer signals beyond immediate neighbors.

A complementary critical insight into this problem was articulated by Wang et al. (2022), who uncovered a paradoxical phenomenon: as GNN depth increases, predictive accuracy diminishes, yet model confidence paradoxically rises, as shown in Figure 1. Highlighting that calibration errors may not merely from local neighbor information, but from multi-scale structural effects spanning across the graph. This observation reveals the core weakness of local-neighbor-based calibration: while shallow cues correlate with uncertainty, they cannot capture the non-local dependencies that drive systematic confidence misalignment in deeper GNNs.

These findings underscore the necessity of calibration approaches that go beyond local neighbor statistics. Prior works (Wang et al., 2022; Hsu et al., 2022; Tang et al., 2024) have emphasized some graph features, such as homophily and node similarity. GETS (Zhuang et al., 2024) partially addresses this by incorporating degree embeddings into a mixture-of-experts framework. While all these methods lack the ability to capture multi-hop structural patterns or incorporate unstable confidence. Therefore, addressing the above limitation and improving the calibration requires leveraging multi-hop context and structure-aware features, enabling the model to better handle both under- and over-confidence in structurally diverse regions of the graph.

## 3.3 WAVELET-AWARE TEMPERATURE SCALING

We propose WATS, a lightweight and effective node-wise calibration framework that can be seamlessly applied to any pretrained GNN with scalability to large graphs. Unlike conventional or graph-specific post-hoc methods that rely on global or one-hop features, WATS introduces a structural perspective by leveraging graph wavelet features with tunable scales.

These wavelet representations capture rich, scalable structural signals (Hammond et al., 2011; Crovella & Kolaczyk, 2003), often neglected in calibration. By learning a temperature for each node based on its structural embedding, WATS aligns confidence with correctness in a fine-grained, node-specific manner. In addition to its strong empirical performance, WATS is also architecture-agnostic, making it broadly applicable across diverse graph types and calibration scenarios.

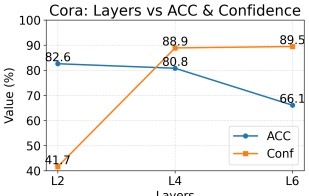 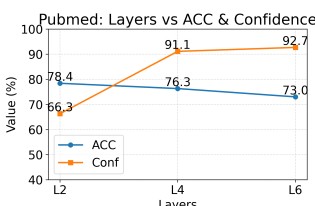 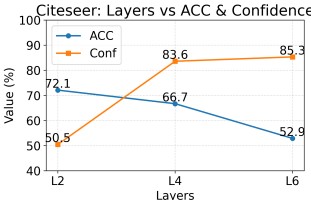

Figure 1: Test accuracy (ACC) and average predictive confidence of GCNs with increasing depth on Cora, Pubmed, and Citeseer. In all three datasets, deeper models exhibit decreasing accuracy while confidence increases, indicating depth-induced miscalibration.

### 3.3.1 GRAPH WAVELET TRANSFORM

Traditional graph signal processing often relies on graph Fourier transform, which projects signals into the spectral domain using the eigenvectors of the normalized graph Laplacian $\mathbf{L}_{\text{sym}} = \mathbf{I} - \mathbf{D}^{-1/2}\mathbf{A}\mathbf{D}^{-1/2}$ as orthonormal bases. Given a signal $\mathbf{x} \in \mathbb{R}^N$, its Fourier transform is defined as $\hat{\mathbf{x}} = \mathbf{U}^\top\mathbf{x}$ and the inverse as $\mathbf{x} = \mathbf{U}\hat{\mathbf{x}}$, where $\mathbf{U}$ contains the eigenvectors of $\mathbf{L}_{\text{sym}}$ (Shuman et al., 2013). While this formulation enables spectral filtering via $\mathbf{U}g_\theta\mathbf{U}^\top\mathbf{x}$, it suffers from several limitations (Hammond et al., 2011; Xu et al., 2019; Zheng et al., 2021): (1) The eigendecomposition of $\mathbf{L}_{\text{sym}}$ has high computational cost ($\mathcal{O}(N^3)$); (2) $\mathbf{U}$ is generally dense, making the transform costly for large graphs; (3) The resulting filters lack localization in the vertex domain, limiting their ability to capture localized structural patterns.

To overcome these issues, we adopt the graph wavelet transform, which retains the spectral benefits of Fourier analysis while introducing localization and sparsity. Graph wavelet bases are constructed using a heat kernel scaling function $g(s\lambda) = e^{-s\lambda}$, where $s > 0$ is a scale parameter controlling the diffusion extent. The wavelet operator is defined as:

$$\mathbf{\Psi}_s = \mathbf{U}\operatorname{diag}(g(s\lambda_1), \ldots, g(s\lambda_N))\,\mathbf{U}^\top \tag{4}$$

where $\lambda_i$ are the eigenvalues of $\mathbf{L}_{\text{sym}}$. The inverse transform uses $g(-s\lambda)$, yielding efficient localized filtering analogous to diffusion. Direct computation of $\mathbf{\Psi}_s$ is still impractical for large graphs. To address this, we adopt the Chebyshev polynomial approximation to avoid explicit eigendecomposition, following (Hammond et al., 2011; Xu et al., 2019). We first rescale $\mathbf{L}_{\text{sym}}$ as: $\hat{\mathbf{L}} = \frac{2}{\lambda_{\max}}\mathbf{L}_{\text{sym}} - \mathbf{I}$, $\lambda_{\max} \approx 2$ and define Chebyshev polynomials $\{\mathbf{T}_k\}_{k=0}^K$ via the recurrence: $\mathbf{T}_0 = \mathbf{X}_0, \mathbf{T}_1 = \hat{\mathbf{L}}\mathbf{X}_0, \mathbf{T}_k = 2\hat{\mathbf{L}}\mathbf{T}_{k-1} - \mathbf{T}_{k-2},$ for $k \geq 2$ where $\mathbf{X}_0$ is the initial input signal.

In our setting, we choose $\mathbf{X}_0$ as the $\log$-degree to preserve structural properties while mitigating skewed degree distributions. Degree encodes a node's connectivity and its potential for information aggregation in message-passing GNNs, and in previous work (Zhuang et al., 2024), it is shown to be correlated with miscalibration.

The wavelet scaling function $g(s\lambda) = e^{-s\lambda}$ is approximated using a $K$-order Chebyshev series:

$$g(s\lambda) \approx \frac{1}{2}c_0 + \sum_{k=1}^K c_k T_k(\lambda)$$

with the Chebyshev coefficients $c_k$ given by:

$$c_k = \frac{2}{\pi}\int_0^\pi \cos(k\theta)g\left(s\frac{\lambda_{\max}}{2}(\cos\theta + 1)\right)d\theta$$

As stated in Hammond et al. (2011), these $c_k$ are computable constants before training. The final wavelet-transformed feature matrix $\mathbf{S}$ is constructed by applying this polynomial filter to the input signal:

$$\mathbf{S} = \frac{1}{2}c_0\mathbf{T}_0 + \sum_{k=1}^K c_k\mathbf{T}_k \tag{5}$$

This is followed by row-wise $\ell_1$ normalization:

$$\mathbf{H}_i = \frac{\mathbf{S}_i}{\|\mathbf{S}_i\|_1}, \quad \forall i \in \{1, \ldots, N\} \tag{6}$$

The hyper-parameter $K$ sets the maximum receptive-field size (i.e., the number of hops considered), while the scale parameter $s$ governs the extent of diffusion. A small $s$ restricts diffusion and thus accentuates local structure, whereas a large $s$ allows more extensive diffusion, leading to stronger smoothing and the integration of broader, long-range context. In practice, selecting appropriate values for $k$ and $s$ enables control over the locality and granularity of the wavelet features. This flexibility is crucial for capturing diverse structural patterns across graphs of varying density and topology.

### 3.3.2 NODE-WISE TEMPERATURE SCALING

Based on the extracted wavelet features, we predict a node-specific temperature parameter to rescale the logits produced by the original GNN. Given the feature matrix $\mathbf{H} \in \mathbb{R}^{N \times (K+1)}$, we employ a two-layer multilayer perceptron (MLP) to capture the non-linear relationship and predict the temperatures:

$$\tau_i = \text{Softplus}(\text{MLP}(\mathbf{H}_i)) \tag{7}$$

where $\mathbf{h}_i$ is the wavelet feature vector for node $i$, and Softplus ensures the positivity of the predicted temperatures. This design provides a flexible and efficient mechanism for uncertainty calibration across the graph. The calibrated logits are obtained via post-hoc temperature scaling:

$$\tilde{z}_i = \frac{z_i}{\tau_i}$$

where $z_i$ is the original output logit from the GNN, and $\tilde{z}_i$ is the rescaled logit after calibration. The temperature predictor is trained by minimizing the cross-entropy loss on the validation set using the rescaled logits.

## 4 EXPERIMENT

### 4.1 EXPERIMENT SETTING

We evaluate the calibration performance of our proposed WATS method on nine widely-used graph datasets: Cora (McCallum et al., 2000), Citeseer (Giles et al., 1998), Pubmed (Sen et al., 2008), Cora-Full (Bojchevski & Günnemann, 2017), Computers (Shchur et al., 2018), Photo (Shchur et al., 2018), Reddit (Hamilton et al., 2017), Roman and Tolokers(Platonov et al., 2023). These datasets cover a range of graph sizes, homophily, and label complexities, providing a comprehensive benchmark for calibration analysis, detailed graph summary is shown in Appendix.

Following previous practice (Wang et al., 2021; Hsu et al., 2022; Tang et al., 2024), we adopt three commonly used GNN architectures as base models, which are GCN (Kipf & Welling, 2016), GAT (Veličković et al., 2017) and GCNII (Chen et al., 2020). The models are trained under a semi-supervised node classification setting. After training, we perform post-hoc calibration using different methods without modifying the model parameters. Detailed training settingS of these based models are shown in Appendix.

Follow the experiment settings (Hsu et al., 2022; Tang et al., 2024; Zhuang et al., 2024), We randomly use 20% of nodes for training, 10% for validation and calibration training, and 70% for testing. For each method, calibration parameters are learned on the validation set and evaluated on the test set. Calibration performance is measured using the ECE with 10 bins.

We compare several post-hoc calibration methods. TS applies a global temperature to all logits (Guo et al., 2017), while ETS averages predictions from multiple temperature-tuned models (Zhang et al., 2020). CaGCN uses a lightweight GCN to learn node-specific temperatures (Wang et al., 2021), and GATS employs attention-based aggregation over one-hop neighbors (Hsu et al., 2022). GETS introduces a sparse mixture-of-experts that combines degree, features, and logits (Zhuang et al., 2024). WATS, our proposed method, predicts temperatures using tunable graph wavelet features and rescale logits. The detailed experiment setting are displayed in detail in Appendix.

### 4.2 EVALUATION AND ANALYSIS

We evaluate the calibration effectiveness of WATS across nine benchmark datasets and three representative GNN architectures, with results summarized in Table 1. Empirical findings demonstrate

Table 1: Each result is reported as the mean ± standard deviation over 10 runs. 'Uncalib' refers to uncalibrated outputs, and 'oom' indicates out-of-memory failures where the method could not complete. Best performance on ECE are highlighted for each configuration.

| Dataset | Model | Uncalib | TS | ETS | CAGCN | GATS | GETS | WATS |
|---|---|---|---|---|---|---|---|---|
| Citeseer | GCN | 23.20 ± 3.21 | 2.57 ± 0.78 | 3.45 ± 1.03 | 4.44 ± 1.47 | 2.38 ± 0.65 | 4.09 ± 1.36 | **2.11 ± 0.43** |
| | GAT | 15.61 ± 1.14 | 3.22 ± 0.29 | 3.55 ± 0.41 | 3.35 ± 0.41 | 3.22 ± 0.24 | 3.80 ± 2.05 | **3.13 ± 0.23** |
| | GCNII | 13.32 ± 10.99 | 7.39 ± 4.38 | 7.43 ± 4.48 | 8.65 ± 1.77 | 8.66 ± 2.62 | **6.68 ± 3.42** | 7.27 ± 3.49 |
| Computers | GCN | 5.94 ± 0.52 | 3.88 ± 0.70 | 3.91 ± 0.49 | 2.04 ± 0.34 | 3.34 ± 0.61 | 2.94 ± 1.26 | **1.20 ± 0.19** |
| | GAT | 5.86 ± 1.26 | 2.12 ± 0.19 | 2.11 ± 0.20 | 2.99 ± 0.64 | **2.01 ± 0.17** | 3.95 ± 3.73 | 2.17 ± 0.16 |
| | GCNII | 10.30 ± 1.37 | 10.30 ± 0.67 | 6.91 ± 0.87 | 5.69 ± 0.47 | 5.62 ± 0.56 | **2.89 ± 1.26** | 3.89 ± 0.68 |
| Cora | GCN | 22.44 ± 1.17 | 2.25 ± 0.33 | 2.20 ± 0.44 | 2.79 ± 0.50 | 2.98 ± 0.59 | 2.96 ± 0.47 | **1.82 ± 0.27** |
| | GAT | 17.26 ± 0.38 | 2.03 ± 0.31 | **1.92 ± 0.31** | 2.56 ± 0.38 | 2.15 ± 0.30 | 2.97 ± 0.47 | 2.02 ± 0.30 |
| | GCNII | 17.35 ± 3.28 | 3.38 ± 0.92 | 3.35 ± 0.93 | 4.35 ± 2.35 | 3.43 ± 1.07 | 6.76 ± 4.94 | **3.23 ± 1.01** |
| Cora-full | GCN | 27.79 ± 0.22 | 5.06 ± 0.10 | 5.00 ± 0.09 | 3.87 ± 0.22 | 5.13 ± 0.10 | 3.11 ± 1.95 | **1.94 ± 0.11** |
| | GAT | 37.21 ± 0.37 | 2.50 ± 0.23 | 1.32 ± 0.16 | 4.79 ± 0.34 | 2.70 ± 0.26 | 2.16 ± 1.11 | **1.11 ± 0.18** |
| | GCNII | 9.66 ± 1.27 | 3.51 ± 0.62 | 3.50 ± 0.61 | 3.28 ± 0.97 | 3.50 ± 0.59 | 3.01 ± 0.88 | **2.92 ± 0.98** |
| Photo | GCN | 3.33 ± 0.22 | 2.45 ± 0.22 | 2.47 ± 0.20 | 1.72 ± 0.22 | 2.22 ± 0.19 | 3.25 ± 1.63 | **1.64 ± 0.31** |
| | GAT | 3.21 ± 0.47 | 1.81 ± 0.43 | 2.34 ± 0.50 | 1.71 ± 0.10 | 1.80 ± 0.43 | 3.05 ± 1.67 | **1.63 ± 0.18** |
| | GCNII | 9.66 ± 1.27 | 1.65 ± 0.54 | 1.88 ± 0.66 | 1.50 ± 0.36 | 1.65 ± 0.56 | 3.15 ± 1.28 | **1.33 ± 0.48** |
| Pubmed | GCN | 14.33 ± 1.20 | 2.55 ± 0.38 | 2.81 ± 0.47 | 1.82 ± 0.36 | 2.30 ± 0.52 | 2.34 ± 0.51 | **1.12 ± 0.09** |
| | GAT | 10.67 ± 0.30 | 0.88 ± 0.09 | 0.88 ± 0.09 | 0.91 ± 0.11 | 0.89 ± 0.10 | 0.90 ± 0.22 | **0.84 ± 0.08** |
| | GCNII | 12.94 ± 1.18 | 3.21 ± 0.91 | 3.65 ± 0.91 | **2.02 ± 1.67** | 2.42 ± 0.93 | 2.23 ± 0.31 | 2.10 ± 0.34 |
| Tolokers | GCN | 3.36 ± 0.12 | 2.99 ± 1.59 | 3.44 ± 0.67 | 2.54 ± 0.38 | 3.67 ± 2.64 | 2.54 ± 1.08 | **2.45 ± 0.22** |
| | GAT | 3.48 ± 0.37 | 3.03 ± 0.21 | 3.10 ± 0.19 | **1.69 ± 0.15** | 2.89 ± 0.27 | 2.51 ± 0.77 | 2.16 ± 0.22 |
| | GCNII | 6.40 ± 0.62 | 4.39 ± 0.61 | 4.02 ± 0.33 | 4.38 ± 0.61 | 4.02 ± 0.33 | 4.41 ± 0.48 | **3.34 ± 0.20** |
| Roman | GCN | 10.25 ± 0.40 | 4.02 ± 0.27 | 4.36 ± 0.35 | 4.67 ± 0.58 | 3.95 ± 0.24 | 4.61 ± 0.36 | **3.42 ± 0.77** |
| | GAT | 16.43 ± 1.66 | 3.91 ± 0.89 | 4.61 ± 0.94 | 4.51 ± 0.68 | 3.62 ± 0.84 | 4.48 ± 1.44 | **3.31 ± 0.59** |
| | GCNII | 21.00 ± 0.42 | 3.61 ± 0.65 | 3.61 ± 0.65 | 4.62 ± 0.96 | 4.38 ± 0.84 | 4.34 ± 1.18 | **2.92 ± 1.27** |
| Reddit | GCN | 6.69 ± 0.12 | 1.64 ± 0.05 | 1.64 ± 0.05 | 1.45 ± 0.08 | oom | 2.20 ± 0.36 | **0.90 ± 0.05** |
| | GAT | 4.79 ± 0.16 | 3.29 ± 0.08 | 3.35 ± 0.12 | 0.73 ± 0.08 | oom | 1.10 ± 0.11 | **0.54 ± 0.08** |
| | GCNII | 17.73 ± 1.10 | 1.41 ± 0.36 | 1.44 ± 0.34 | 1.20 ± 1.20 | oom | 2.99 ± 0.53 | **0.88 ± 0.28** |

that WATS consistently achieves the lowest ECE in most of scenarios, highlighting its efficacy in leveraging localized, flexibly scaled structural information for post-hoc uncertainty calibration. For architectures like GCNII, While their initial residual connections effectively mitigate over-smoothing, the strong connections force the model to rely heavily on the original node features. In contrast, our results prove that graph wavelets are able to capture sufficient local topology information to correct these confidence levels, effectively addressing the limitations of the base models. Beyond achieving superior average ECE scores, WATS also exhibits reduced standard deviations across runs, indicating improved robustness and stability compared to existing methods. These evidence prove that graph wavelet is able to capture sufficient local topology information to correct the confidence level. Moreover, even when the base model is already reasonably well calibrated, for example, on the Photo and Computers, WATS consistently delivers further reductions in calibration error, demonstrating its ability to adaptively refine predictive confidence across a range of baseline reliability levels.

To illustrate this effect, we visualize WATS on Citeseer in Figure 2. The reliability diagram in Figure 2a shows that the uncalibrated model is systematically under confident, with predicted probabilities below empirical accuracy across bins. After calibration, the curve aligns closely with the diagonal, indicating improved confidence–accuracy agreement. The degree stratified analysis in Figure 2b shows that under confidence is strongest for nodes with low degree; calibration restores agreement across all degree ranges and reduces variance. Overall, WATS improves calibration and robustness, especially in structurally sparse regions. Full visualizations for the main experiments are provided in the Appendix.

Furthermore, GATS's reliance on full attention over a node's neighborhood leads to poor memory scalability and resulting in out-of-memory failures on large graphs such as Reddit, while WATS remains efficient, which improve the scalability of WATS.

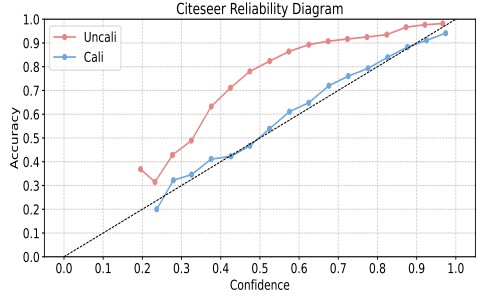 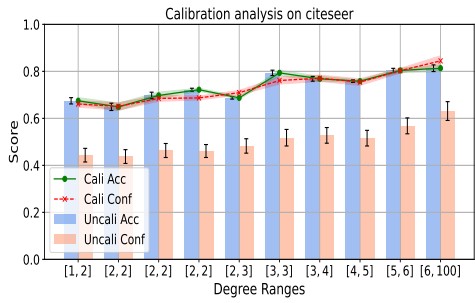

(a) Reliability diagram on citeseer.        (b) Degree-binned calibration analysis on citeseer.

Figure 2: "Uncali" refers to the uncalibrated result and "Cali" refers to the calibrated result. (a) shows the reliability diagram comparing calibrated and uncalibrated outputs. The diagonal dashed line indicates perfect calibration (b) presents a degree-binned analysis of accuracy and confidence. Solid and dashed lines represent calibrated accuracy and confidence respectively.

## 4.3 ABLATION STUDY

### 4.3.1 DIFFERENT BASE SIGNAL

We ablate the base signal used by the WATS with log-degree, raw degree, and identity metrix on GCN, while keeping all other components and hyperparameters fixed. Table 2 reports ECE ($\downarrow$) on nine datasets. Log-degree attains the best or tied-best ECE on the majority of datasets, consistently improving over raw degree and matching or approaching the identity baseline. The only exception is Pubmed, where the identity feature yields a marginally lower ECE. These results support the use of a logarithmic transform to compress extreme degrees while preserving the connectivity ordering, thereby stabilizing learning and improving generalization from low-degree to high-degree regions.

Table 2: ECE ($\downarrow$) comparison between log-degree, raw degree, and an identity matrix as base signal. This comparison isolates the effect of the base structural signal used by the temperature regressor.

|  | Citeseer | Computers | Cora | Photo | Cora-Full | Pubmed | Reddit | Roman | Tolokers |
|---|---|---|---|---|---|---|---|---|---|
| Log-degree | **2.11 ± 0.43** | **1.20 ± 0.19** | **1.82 ± 0.27** | **1.41 ± 0.31** | **1.94 ± 0.11** | 1.12 ± 0.09 | **0.90 ± 0.05** | **3.42 ± 0.77** | **2.45 ± 0.22** |
| Degree | 2.13 ± 0.49 | 1.42 ± 0.24 | 2.25 ± 1.00 | 1.81 ± 0.15 | 3.77 ± 0.41 | 1.12 ± 0.17 | 1.09 ± 0.06 | 3.99 ± 0.90 | 2.47 ± 0.22 |
| Identity Matrix | 2.16 ± 0.47 | 1.31 ± 0.20 | 2.21 ± 0.83 | 1.73 ± 0.17 | 2.99 ± 0.39 | **1.08 ± 0.12** | 1.20 ± 0.08 | 3.60 ± 1.07 | 2.73 ± 0.18 |

### 4.3.2 DIFFERENT GRAPH FEATURES

To assess the effectiveness of graph wavelet features in post-hoc calibration, we conduct a comparative analysis against several widely used structural descriptors, including log-degree, betweenness centrality, clustering coefficient, and their various combinations, all evaluated under a consistent GCN-based framework. As summarized in Table 3, wavelet-based representations consistently yield superior calibration performance across most datasets. While certain individual features or their combinations may perform competitively on specific datasets, they tend to exhibit limited generalizability and often result in higher calibration error overall. This highlights the insufficiency of isolated structural indicators and underscores the necessity of incorporating rich, multiscale topological signals. In contrast, graph wavelet features demonstrate both effectiveness and robustness across diverse graph structures, suggesting that the information they encode captures nuanced patterns that cannot be fully replicated by aggregating conventional structural features.

### 4.3.3 SENSITIVITY ANALYSIS OF GRAPH WAVELET HYPER-PARAMETERS.

To assess the robustness of WATS, we perform an exhaustive grid search over the Chebyshev order $k \in \{2, 3, 4\}$ and the heat-kernel scale $s \in \{0.4, 0.8, 1.2, 1.6, 2.0, 2.5, 3.0, 4.0\}$ on nine node-classification benchmarks, we visualize the changes of ECE for varying $k$ and $s$ for Cora-full, Cora, Computers and Roman on Figure 3 (full results about hyperparameters are in Appendix).

Table 3: ECE ($\downarrow$) comparison between graph wavelet and alternative structural features, where "Deg" denote log transformed degree, "Cen" denote betweenness centrality, "Clus" denote clustering coefficient, and 'oom' indicates out-of-memory failures where the method could not complete. Graph wavelet consistently outperforms other variants across most datasets.

| Dataset | Graph wavelet | Deg | Cen | Clus | Deg, Cen | Cen, Clus | Deg, Clus | Deg, Clus, Cen |
|---|---|---|---|---|---|---|---|---|
| Citeseer | **2.11 ± 0.43** | 3.53 ± 1.16 | 3.10 ± 1.05 | 6.75 ± 1.44 | 7.24 ± 1.80 | 7.11 ± 1.80 | 7.10 ± 1.80 | 7.12 ± 1.69 |
| Computers | **1.20 ± 0.19** | 1.61 ± 0.33 | 3.51 ± 0.83 | 2.75 ± 0.82 | 1.82 ± 0.22 | 2.78 ± 0.69 | 2.60 ± 0.71 | 2.72 ± 0.66 |
| Cora | **1.82 ± 0.27** | 2.42 ± 0.72 | 1.86 ± 0.32 | 4.77 ± 0.40 | 4.43 ± 0.63 | 4.51 ± 0.48 | 4.51 ± 0.48 | 4.55 ± 0.40 |
| Cora-full | **1.94 ± 0.11** | 3.08 ± 1.34 | 5.32 ± 0.18 | 5.66 ± 0.35 | 5.16 ± 0.23 | 5.19 ± 0.26 | 5.19 ± 0.26 | 5.18 ± 0.25 |
| Photo | 1.41 ± 0.31 | **1.32 ± 0.29** | 2.23 ± 0.40 | 1.85 ± 0.38 | 1.96 ± 0.30 | 1.93 ± 0.36 | 1.87 ± 0.34 | 1.89 ± 0.35 |
| Pubmed | **1.12 ± 0.09** | 1.40 ± 0.35 | 2.90 ± 0.38 | 2.30 ± 0.29 | 1.83 ± 0.20 | 1.92 ± 0.22 | 1.93 ± 0.22 | 1.94 ± 0.20 |
| Tolokers | 2.45 ± 0.22 | 2.35 ± 0.34 | 3.02 ± 1.46 | 2.79 ± 0.34 | **2.29 ± 0.39** | 2.71 ± 0.27 | 3.63 ± 0.88 | 2.58 ± 0.21 |
| Roman | **3.42 ± 0.77** | 4.27 ± 0.68 | 4.04 ± 0.50 | 3.88 ± 0.65 | 3.60 ± 0.77 | 3.71 ± 0.78 | 4.03 ± 0.39 | 3.89 ± 0.58 |
| Reddit | **0.90 ± 0.05** | 1.58 ± 0.17 | oom | oom | oom | oom | oom | oom |

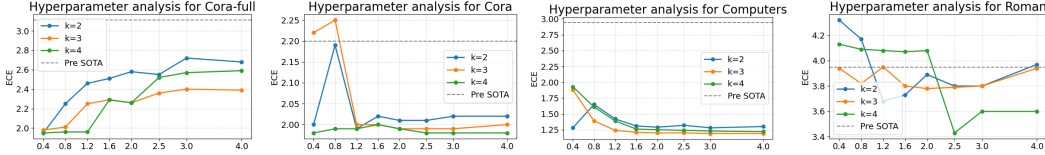

Figure 3: Sensitivity analysis of wavelet hyper-parameters. Each plot shows the ECE scores on different datasets with varying wavelet scale parameter $s$ (x-axis) and polynomial order $k$. Each line represents a different Chebyshev order $k$: blue for $k = 2$, orange for $k = 3$, green for $k = 4$ and grey for previous SOTA.

On highly homophilous graphs, WATS exhibits reduced hyperparameter sensitivity: variations in the Chebyshev order $k$ and the diffusion scale $s$ induce only minor changes in ECE, yielding flatter performance curves and stronger robustness when $s > 1.2$ on high-homophily graphs, such as Cora and Computers. In contrast, on low-homophily graphs, like Cora-full, calibration is more sensitive to $s$ and $k$. On heterophilous graphs, calibration quality depends on both $k$ and $s$: small orders constrain the receptive field and miss meso-scale structure, whereas large orders amplify noise propagation and degrade calibration. These observations yield practical guidance that favors a moderate spectral scale and a mid-level polynomial order. A sensible default for new graphs is $k = 3$ with $s = 2.0$.

However, WATS surpasses the previous SOTA across a broad and practical range even away from the optimum; for example, $k \in \{3, 4\}$ with a moderate $s$ already delivers consistently lower ECE, underscoring the robustness and generality of wavelet-based structural signals for calibration.

### 4.4 COMPLEXITY ANALYSIS

We further compare the complexity with other post-hoc calibration method to prove the computational efficiency. Our method consists of two main components: graph wavelet feature extraction and a two-layer MLP for temperature prediction. Let $K$ be the Chebyshev polynomial order. Each Chebyshev term requires a sparse matrix multiplication, leading to a total time complexity of $\mathcal{O}(K|\mathcal{E}| + |\mathcal{V}|K)$, where $|\mathcal{E}|$ and $|\mathcal{V}|$ denote the number of edges and nodes, respectively. The first term accounts for $K$ sparse multiplications over the Laplacian, while the second accounts for the intermediate tensor concatenation and normalization steps. The wavelet features of each node (dimension $K + 1$) are passed through a two-layer MLP with hidden size $h$. The per-node computation costs $\mathcal{O}((K + 1)h)$, and thus the total cost over all nodes is: $\mathcal{O}(|\mathcal{V}|kh)$. Combining the above, the overall time complexity of our method is:

$$\mathcal{O}(k|\mathcal{E}| + |\mathcal{V}|k + |\mathcal{V}|kh) = \mathcal{O}(k|\mathcal{E}| + |\mathcal{V}|kh).$$

Compared to CaGCN (Wang et al., 2021) with $\mathcal{O}(|\mathcal{E}|F + |\mathcal{V}|F^2)$, and GATS (Veličković et al., 2017) with $\mathcal{O}(|\mathcal{E}|FH + |\mathcal{V}|F^2)$ where $F$ stands for the dimension of the node hidden features and $H$ is the number of independent attention heads, our model is significantly more efficient, especially when $F$ is large or multi-head attention is used. GETS (Zhuang et al., 2024) incurs higher cost due to expert selectionmechanism, which introduces additional complexity by selecting the top-$k$ experts per node. This results in an overall complexity of $\mathcal{O}(k(|\mathcal{E}|F + |\mathcal{V}|F^2 + |\mathcal{V}|MF)$, where $k \ll M$.

In practice, the wavelet transformation can be precomputed and reused as a static input. The wall-clock time and memory usage of WATS and other baseline methods across graph datasets of varying complexity are reported in the Appendix.

## 5    LIMITATIONS AND FUTURE WORK

**Limitations:** The scope of our current study is primarily confined to node classification. Furthermore, WATS relies on the assumption that topological signals inherently correlate with model logits. In scenarios where this correlation is weak or spurious, relying on wavelet-derived temperatures could potentially degrade calibration reliability

**Future work:** We aim to explore the integration of structurally similar, albeit spatially distant, neighborhoods to incorporate a broader global structural context. Specifically, this entails capturing and aggregating nodes that share similar structural embeddings regardless of their multi-hop distance. We hypothesize that this approach could further enhance both calibration performance and model robustness, provided that the incorporation of global information is carefully regularized to avoid introducing extraneous noise. Additionally, we plan to investigate the generalizability of graph wavelets in improving calibration and predictive performance across a wider array of graph-learning tasks, including edge prediction and dynamic graph modeling.

## 6    CONCLUSION

We introduce WATS, a lightweight post-hoc calibration framework that assigns node-specific temperatures from graph wavelet features. By leveraging structural representations, WATS captures diverse structural patterns and implicitly broadens each node's receptive field, improving post-hoc information use with minimal overhead. Across nine benchmarks and three GNN backbones, WATS consistently attains the lowest ECE and markedly stabilizes calibration, yielding more reliable predictions, especially in high-risk settings.

## 7    ACKNOWLEDGEMENT

This work was supported by Sichuan Science and Technology Program 2025YFHZ0162.

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

## A    USAGE OF LARGE LANGUAGE MODEL

In writing this paper, we used a Large Language Model (LLM) solely as a writing assistant to enhance linguistic quality, not to generate substantive content. The LLM was employed to improve readability, concision, and academic tone; to correct grammar, spelling, and punctuation; and to strengthen logical flow and transitions throughout the text.

## B    EXPERIMENT SETTING

We randomly conduct the train test split 10 times for each dataset with identical random seed. We employed the GATS, GETS and CaGCN based on their paper and code. The hyperparameters for backbone GNNs training are based on the complexity of graph data. The detail is given below Table 4 and 5.

Table 4: Summary of training parameters of GCN and GAT

| Dataset | Hidden Dim. | Dropout | Epochs | Learning Rate | Weight Decay |
|---------|-------------|---------|--------|---------------|--------------|
| Citeseer | 16 | 0.5 | 200 | $1 \times 10^{-2}$ | $5 \times 10^{-4}$ |
| Computers | 64 | 0.8 | 200 | $1 \times 10^{-2}$ | $1 \times 10^{-3}$ |
| Cora-full | 64 | 0.8 | 200 | $1 \times 10^{-2}$ | $1 \times 10^{-3}$ |
| Cora | 16 | 0.5 | 200 | $1 \times 10^{-2}$ | $5 \times 10^{-4}$ |
| Photo | 64 | 0.8 | 200 | $1 \times 10^{-2}$ | $1 \times 10^{-3}$ |
| Pubmed | 16 | 0.5 | 200 | $1 \times 10^{-2}$ | $5 \times 10^{-4}$ |
| Tolokers | 16 | 0.2 | 200 | $1 \times 10^{-2}$ | $5 \times 10^{-4}$ |
| Roman | 16 | 0.2 | 200 | $1 \times 10^{-2}$ | $5 \times 10^{-4}$ |
| Reddit | 16 | 0.5 | 200 | $1 \times 10^{-2}$ | $5 \times 10^{-4}$ |

Table 5: Summary of training parameters of GCNII

| Dataset | Layers | Hidden Dim. | Dropout | Epochs | Learning Rate | Weight Decay |
|---------|--------|-------------|---------|--------|---------------|--------------|
| Citeseer | 16 | 16 | 0.5 | 200 | $1 \times 10^{-2}$ | $5 \times 10^{-4}$ |
| Computers | 16 | 64 | 0.8 | 200 | $1 \times 10^{-2}$ | $1 \times 10^{-3}$ |
| Cora-full | 16 | 64 | 0.8 | 200 | $1 \times 10^{-2}$ | $1 \times 10^{-3}$ |
| Cora | 16 | 16 | 0.5 | 200 | $1 \times 10^{-2}$ | $5 \times 10^{-4}$ |
| Photo | 16 | 64 | 0.8 | 200 | $1 \times 10^{-2}$ | $1 \times 10^{-3}$ |
| Pubmed | 16 | 16 | 0.5 | 200 | $1 \times 10^{-2}$ | $5 \times 10^{-4}$ |
| Tolokers | 16 | 16 | 0.2 | 200 | $1 \times 10^{-2}$ | $5 \times 10^{-4}$ |
| Roman | 16 | 16 | 0.2 | 200 | $1 \times 10^{-2}$ | $5 \times 10^{-4}$ |
| Reddit | 16 | 16 | 0.5 | 200 | $1 \times 10^{-2}$ | $5 \times 10^{-4}$ |

Full details of WATS in calibration is given be on Table 6. Hidden dimension and drop out are chosen based on the data complexity. The hyperparameter of graph wavelet $k$ and $s$ are chosen based on the Ablation study.

Full details of the chosen datasets is given on Table 7. It reports the number of nodes, edges, average node degree, input feature dimensions, and number of classes for each dataset. These datasets cover a diverse range of graph sizes, densities, and classification tasks. This diversity ensures a comprehensive evaluation of the proposed method under varying structural and semantic conditions.

**Computational Environment.**    All experiments are conducted using the following environment with PyTorch 2.4.0 (Python 3.11, CUDA 12.4.1), Hardware: NVIDIA RTX 4090 GPU with 32 GB RAM on Runpod cloud service (Ubuntu 22.04)

Table 6: Calibration settings for WATS

| Dataset | Hidden Dim. | Dropout | k | s |
|---------|-------------|---------|---|---|
| Citeseer | 32 | 0.4 | 3 | 0.8 |
| Computers | 64 | 0.4 | 3 | 2.5 |
| Cora-full | 128 | 0.2 | 4 | 0.4 |
| Cora | 16 | 0.95 | 4 | 0.4 |
| Photo | 32 | 0.4 | 4 | 0.8 |
| Pubmed | 32 | 0.4 | 4 | 0.4 |
| Tolokers | 32 | 0.6 | 4 | 3 |
| Roman | 32 | 0.2 | 4 | 2.5 |
| Reddit | 64 | 0.4 | 4 | 3 |

Table 7: Summary of selected datasets

| Dataset | #Nodes | #Edges | Avg. Degree | #Features | #Classes |
|---------|--------|--------|-------------|-----------|----------|
| Citeseer | 3,327 | 12,431 | 7.4 | 3,703 | 6 |
| Computers | 13,381 | 491,556 | 73.4 | 767 | 10 |
| Cora | 2,708 | 13,264 | 9.7 | 1,433 | 7 |
| Cora-full | 18,800 | 144,170 | 15.3 | 8,710 | 70 |
| Photo | 7,487 | 238,087 | 63.6 | 745 | 8 |
| Pubmed | 19,717 | 108,365 | 10.9 | 500 | 3 |
| Tolokers | 11,758 | 1,038,000 | 176.6 | 10 | 2 |
| Roman | 22,662 | 65,854 | 5.8 | 300 | 18 |
| Reddit | 232,965 | 114,848,857 | 98.5 | 602 | 41 |

## C    TIME AND PEAK MEMORY USAGE

We report the wall-clock time and memory usage of WATS and other baseline methods across graph datasets of varying complexity in Table 8 and Table 9, respectively.

Table 8: Comparison of Computation Time (seconds)

| Dataset | WATS Feat. Time (s) | WATS Calib Time (s) | GETS Calib Time (s) | GATS Calib Time (s) | TS Calib Time (s) | CaGCN Calib Time (s) |
|---------|---------|---------|---------|---------|---------|---------|
| **Cora** | 0.0624 | 1.0134 | 22.2517 | 2.1201 | 0.3941 | 4.1283 |
| **Computer** | 0.2374 | 0.7716 | 10.8242 | 5.0126 | 0.2408 | 4.9372 |
| **Cora-Full** | 0.3134 | 1.0461 | 9.2106 | 5.2835 | 0.4754 | 4.7652 |
| **Reddit** | 42.6636 | 1.3149 | 45.2529 | NAN | 1.1054 | 20.8815 |

Table 9: Comparison of Memory Usage

| Dataset | WATS Feat. Memory | WATS Calib Memory | GETS Calib Memory | GATS Calib Memory | TS Calib Memory | CaGCN Calib Memory |
|---------|---------|---------|---------|---------|---------|---------|
| **Cora** | 207.31 MB | 207.92 MB | 97.39 MB | 95.00 MB | 94.65 MB | 94.65 MB |
| **Computer** | 207.83 MB | 211.62 MB | 195.91 MB | 313.41 MB | 172.45 MB | 172.46 MB |
| **Cora-Full** | 1955.27 MB | 1963.90 MB | 1401.74 MB | 1349.16 MB | 1344.47 MB | 1343.71 MB |
| **Reddit** | 12456.00 MB | 4322.00 MB | 5938.96 MB | >17.54 GiB | 3855.37 MB | 3857.81 MB |

## D    HYPERPARAMETER ANALYSIS RESULTS

Here is the full result for the experiment on hyperparameter analysis. Tables 10 to 12 report the calibration performance measured by ECE of WATS under varying graph wavelet hyperparameters, specif-

ically the Chebyshev order $k \in \{2, 3, 4\}$ and diffusion scale $s \in \{0.4, 0.8, 1.2, 1.6, 2.0, 2.5, 3.0, 4.0\}$. For each dataset, ECE values are presented across a range of $s$ values.

Table 10: ECE ($\downarrow$) for different diffusion scales $s$ with Chebyshev order $K=2$.

|  | $s=0.4$ | $s=0.8$ | $s=1.2$ | $s=1.6$ | $s=2.0$ | $s=2.5$ | $s=3$ | $s=4$ |
|---|---|---|---|---|---|---|---|---|
| Citeseer | $2.54 \pm 0.86$ | $2.50 \pm 0.86$ | $2.46 \pm 0.88$ | $2.47 \pm 0.89$ | $2.47 \pm 0.91$ | $2.48 \pm 0.92$ | $2.50 \pm 0.93$ | $2.51 \pm 0.94$ |
| Cora | $2.00 \pm 0.26$ | $1.99 \pm 0.26$ | $1.99 \pm 0.27$ | $2.02 \pm 0.27$ | $2.01 \pm 0.28$ | $2.01 \pm 0.27$ | $2.02 \pm 0.28$ | $2.02 \pm 0.28$ |
| Computers | $2.18 \pm 0.47$ | $1.65 \pm 0.42$ | $1.42 \pm 0.30$ | $1.48 \pm 0.27$ | $1.37 \pm 0.26$ | $1.32 \pm 0.22$ | $1.53 \pm 0.37$ | $1.50 \pm 0.35$ |
| Pubmed | $1.17 \pm 0.19$ | $1.19 \pm 0.19$ | $1.20 \pm 0.20$ | $1.20 \pm 0.17$ | $1.21 \pm 0.18$ | $1.20 \pm 0.17$ | $1.18 \pm 0.13$ | $1.17 \pm 0.13$ |
| Reddit | $1.19 \pm 0.09$ | $1.15 \pm 0.09$ | $1.10 \pm 0.11$ | $1.08 \pm 0.12$ | $1.07 \pm 0.13$ | $1.03 \pm 0.12$ | $0.99 \pm 0.09$ | $0.96 \pm 0.11$ |
| Cora-full | $1.95 \pm 0.21$ | $2.25 \pm 0.25$ | $2.46 \pm 0.24$ | $2.51 \pm 0.29$ | $2.58 \pm 0.35$ | $2.55 \pm 0.37$ | $2.72 \pm 0.33$ | $2.68 \pm 0.41$ |
| Photo | $1.75 \pm 0.23$ | $1.76 \pm 0.22$ | $1.71 \pm 0.24$ | $1.72 \pm 0.22$ | $1.70 \pm 0.25$ | $1.71 \pm 0.22$ | $1.69 \pm 0.22$ | $1.71 \pm 0.23$ |
| Roman | $4.32 \pm 0.70$ | $4.17 \pm 0.83$ | $3.68 \pm 0.68$ | $3.73 \pm 0.70$ | $3.89 \pm 0.63$ | $3.80 \pm 0.57$ | $3.80 \pm 0.56$ | $3.97 \pm 0.36$ |
| Tolokers | $2.71 \pm 0.13$ | $2.71 \pm 0.17$ | $2.66 \pm 0.17$ | $2.73 \pm 0.13$ | $2.70 \pm 0.17$ | $2.76 \pm 0.17$ | $2.70 \pm 0.21$ | $2.74 \pm 0.16$ |

Table 11: ECE ($\downarrow$) for different diffusion scales $s$ with Chebyshev order $K=3$.

|  | $s=0.4$ | $s=0.8$ | $s=1.2$ | $s=1.6$ | $s=2.0$ | $s=2.5$ | $s=3$ | $s=4$ |
|---|---|---|---|---|---|---|---|---|
| Citeseer | $2.21 \pm 0.49$ | $2.11 \pm 0.43$ | $2.18 \pm 0.52$ | $2.25 \pm 0.59$ | $2.25 \pm 0.61$ | $2.24 \pm 0.62$ | $2.25 \pm 0.61$ | $2.30 \pm 0.65$ |
| Cora | $2.22 \pm 0.87$ | $2.25 \pm 1.00$ | $2.00 \pm 0.28$ | $2.00 \pm 0.26$ | $1.99 \pm 0.24$ | $1.99 \pm 0.24$ | $1.99 \pm 0.24$ | $2.00 \pm 0.26$ |
| Computers | $1.88 \pm 0.39$ | $1.45 \pm 0.24$ | $1.39 \pm 0.20$ | $1.30 \pm 0.20$ | $1.25 \pm 0.19$ | $1.20 \pm 0.19$ | $1.25 \pm 0.18$ | $1.26 \pm 0.20$ |
| Pubmed | $1.14 \pm 0.18$ | $1.16 \pm 0.18$ | $1.18 \pm 0.18$ | $1.17 \pm 0.12$ | $1.18 \pm 0.13$ | $1.18 \pm 0.13$ | $1.18 \pm 0.12$ | $1.19 \pm 0.12$ |
| Reddit | $1.18 \pm 0.08$ | $1.21 \pm 0.07$ | $1.22 \pm 0.12$ | $1.10 \pm 0.15$ | $0.97 \pm 0.09$ | $0.96 \pm 0.10$ | $0.93 \pm 0.06$ | $0.91 \pm 0.07$ |
| Cora-full | $1.98 \pm 0.20$ | $2.14 \pm 0.24$ | $2.25 \pm 0.14$ | $2.33 \pm 0.17$ | $2.26 \pm 0.17$ | $2.36 \pm 0.33$ | $2.24 \pm 0.16$ | $2.32 \pm 0.31$ |
| Photo | $1.79 \pm 0.20$ | $1.79 \pm 0.22$ | $1.72 \pm 0.29$ | $1.81 \pm 0.18$ | $1.79 \pm 0.20$ | $1.70 \pm 0.30$ | $1.79 \pm 0.22$ | $1.69 \pm 0.28$ |
| Roman | $3.94 \pm 0.16$ | $3.82 \pm 0.46$ | $3.95 \pm 0.17$ | $3.80 \pm 0.51$ | $3.78 \pm 0.54$ | $3.79 \pm 0.53$ | $3.80 \pm 0.49$ | $3.94 \pm 0.98$ |
| Tolokers | $2.67 \pm 0.12$ | $2.67 \pm 0.16$ | $2.76 \pm 0.12$ | $2.71 \pm 0.21$ | $2.64 \pm 0.22$ | $2.55 \pm 0.25$ | $2.50 \pm 0.27$ | $2.66 \pm 0.16$ |

Table 12: ECE ($\downarrow$) for different diffusion scales $s$ with Chebyshev order $K=4$.

|  | $s=0.4$ | $s=0.8$ | $s=1.2$ | $s=1.6$ | $s=2.0$ | $s=2.5$ | $s=3$ | $s=4$ |
|---|---|---|---|---|---|---|---|---|
| Citeseer | $2.77 \pm 1.03$ | $2.68 \pm 1.02$ | $2.63 \pm 1.05$ | $2.56 \pm 0.96$ | $2.73 \pm 1.03$ | $2.73 \pm 1.02$ | $2.74 \pm 1.04$ | $2.74 \pm 1.02$ |
| Cora | $1.98 \pm 0.26$ | $1.99 \pm 0.26$ | $1.99 \pm 0.26$ | $2.00 \pm 0.26$ | $1.99 \pm 0.25$ | $1.98 \pm 0.25$ | $1.98 \pm 0.26$ | $1.98 \pm 0.26$ |
| Computers | $1.93 \pm 0.50$ | $1.44 \pm 0.25$ | $1.30 \pm 0.20$ | $1.28 \pm 0.25$ | $1.23 \pm 0.21$ | $1.30 \pm 0.25$ | $1.26 \pm 0.22$ | $1.29 \pm 0.22$ |
| Pubmed | $1.12 \pm 0.09$ | $1.18 \pm 0.12$ | $1.18 \pm 0.12$ | $1.19 \pm 0.13$ | $1.21 \pm 0.13$ | $1.19 \pm 0.12$ | $1.19 \pm 0.09$ | $1.20 \pm 0.11$ |
| Reddit | $1.17 \pm 0.06$ | $1.19 \pm 0.11$ | $1.18 \pm 0.12$ | $1.13 \pm 0.18$ | $1.02 \pm 0.17$ | $0.93 \pm 0.16$ | $0.90 \pm 0.05$ | $0.91 \pm 0.14$ |
| Cora-full | $1.94 \pm 0.11$ | $2.03 \pm 0.19$ | $2.18 \pm 0.23$ | $2.29 \pm 0.23$ | $2.26 \pm 0.22$ | $2.23 \pm 0.23$ | $2.21 \pm 0.21$ | $2.22 \pm 0.22$ |
| Photo | $1.74 \pm 0.17$ | $1.64 \pm 0.20$ | $1.68 \pm 0.29$ | $1.69 \pm 0.21$ | $1.72 \pm 0.22$ | $1.67 \pm 0.24$ | $1.68 \pm 0.34$ | $1.67 \pm 0.25$ |
| Roman | $4.13 \pm 0.30$ | $4.09 \pm 0.26$ | $4.08 \pm 0.25$ | $4.07 \pm 0.21$ | $4.08 \pm 0.20$ | $3.43 \pm 0.68$ | $3.60 \pm 0.70$ | $3.60 \pm 0.69$ |
| Tolokers | $2.67 \pm 0.17$ | $2.74 \pm 0.18$ | $2.69 \pm 0.22$ | $2.78 \pm 0.22$ | $2.72 \pm 0.27$ | $2.51 \pm 0.22$ | $2.45 \pm 0.22$ | $2.50 \pm 0.23$ |

# E  FULL WATS VISUALIZATIONS

We provide the full visualizations of the calibration performance for WATS. Figures 4 to 11 illustrate the calibration performance of WATS on the other datasets. Each figure includes (a) a reliability diagram showing the alignment between predicted confidence and actual accuracy, and (b) a degree-binned analysis comparing confidence and accuracy before and after calibration. Results show that WATS significantly improves calibration and reduces the discrepancy between accuracy and confidence across all degree ranges and all confidence levels. Error bars indicate standard deviation over 10 runs.

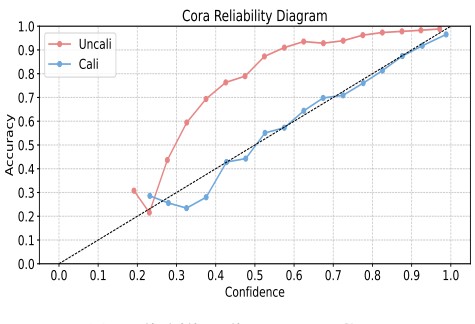

(a) Reliability diagram on Cora.

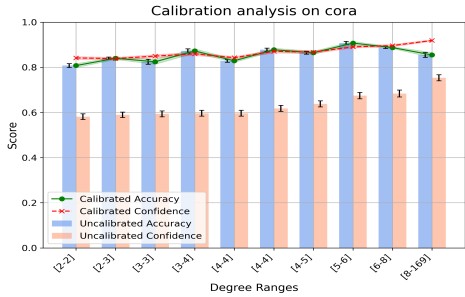

(b) Degree-binned calibration analysis on Cora.

Figure 4: Calibration performance of Cora dataset.

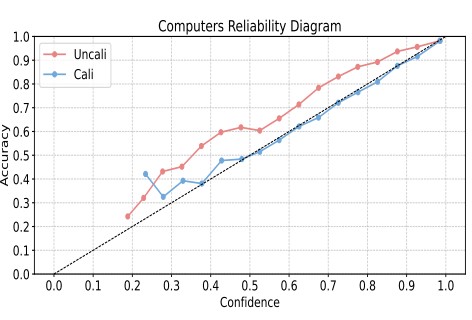

(a) Reliability diagram on Computers.

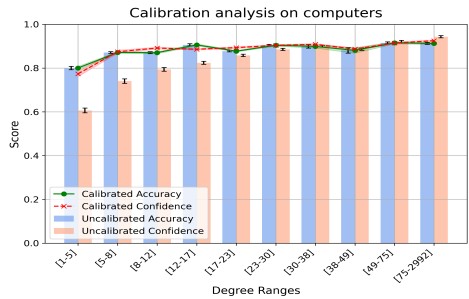

(b) Degree-binned calibration analysis on Computers.

Figure 5: Calibration performance of Computers dataset.

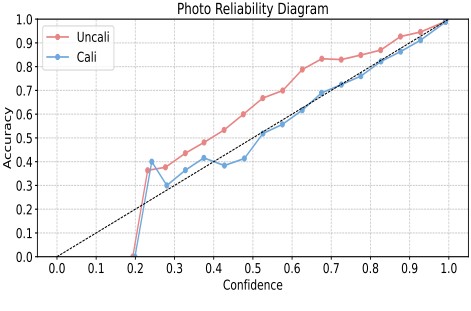

(a) Reliability diagram on Photo.

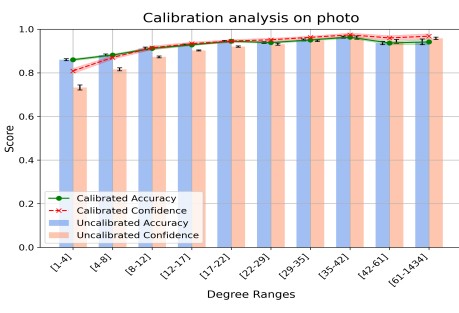

(b) Degree-binned calibration analysis on Photo.

Figure 6: Calibration performance of Photo dataset.

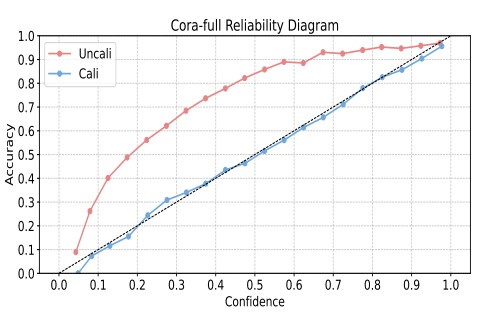

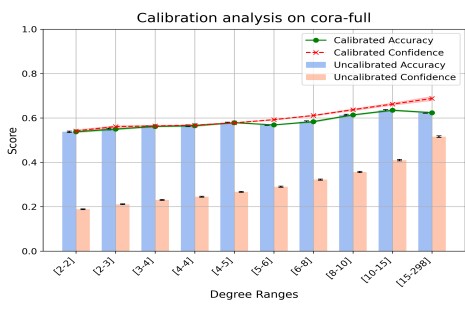

(a) Reliability diagram on Cora-full.

(b) Degree-binned calibration analysis on Cora-full.

Figure 7: Calibration performance of Cora-full dataset.

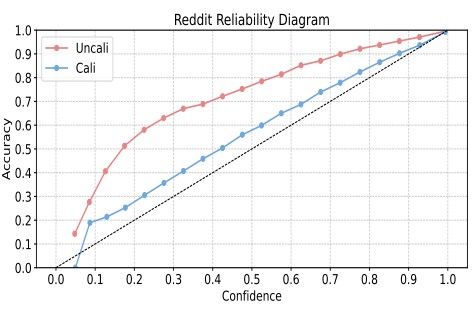

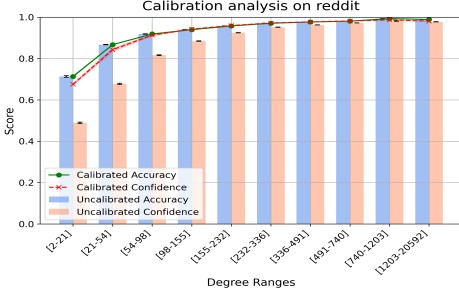

(a) Reliability diagram on Reddit.

(b) Degree-binned calibration analysis on Reddit.

Figure 8: Calibration performance of Reddit dataset.

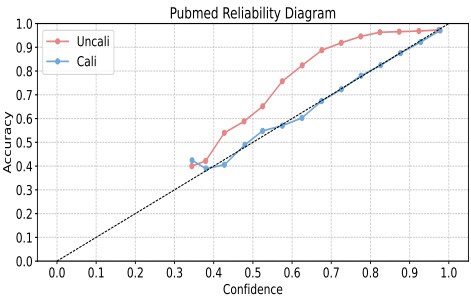

(a) Reliability diagram on Pubmed.

(b) Degree-binned calibration analysis on Pubmed.

Figure 9: Calibration performance of Pubmed dataset.

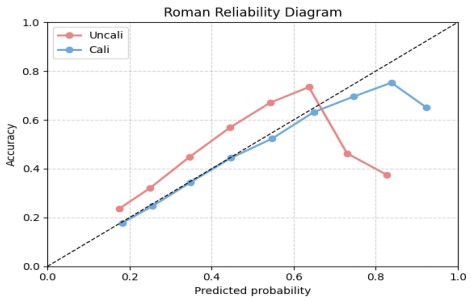
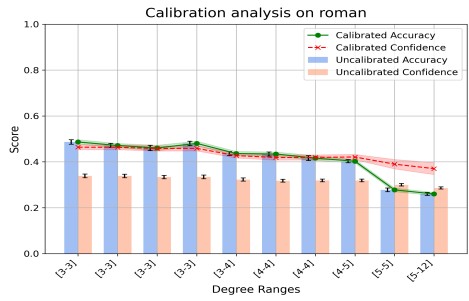

(a) Reliability diagram on Roman.

(b) Degree-binned calibration analysis on Roman.

Figure 10: Calibration performance of Roman dataset.

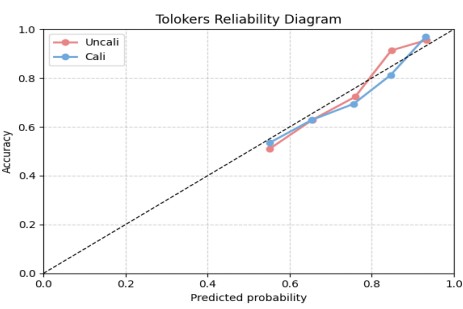
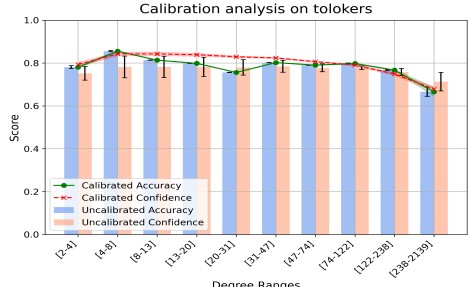

(a) Reliability diagram on Tolokers.

(b) Degree-binned calibration analysis on Tolokers.

Figure 11: Calibration performance of Tolokers dataset.

