# OpenReview forum: "WATS: Wavelet-Aware Temperature Scaling for Reliable Graph Neural Networks"
_ICLR.cc/2026/Conference — ICLR 2026 Poster_

### Official Review · Reviewer_Wzpk · 2025-10-31

**Soundness:** 3
**Presentation:** 3
**Contribution:** 3
**Rating:** 6
**Confidence:** 5

**Summary:**

WATS proposes a post-hoc, label-free calibration method for GNNs using graph wavelet features. A small MLP predicts a per-node temperature, guided by local spectral characteristics computed via heat-kernel wavelets. This allows finer calibration than global temperature scaling and mitigates overconfidence in deep GNNs.

**Strengths:**

1. Simple yet effective node-wise temperature scaling without retraining the base GNN.
2. The wavelet embedding captures multi-scale structural cues for confidence correction.
3. Extensive experiments across datasets and backbones with consistent ECE reduction.
4. Clear complexity and sensitivity analysis, including practical hyperparameter guidance.

**Weaknesses:**

1. Relies on the correlation between structure and calibration error, which may not hold for heterophilous graphs.
2. The wavelet scale parameter tuning could be expensive for a large graph

**Questions:**

1. How robust is WATS to heterophily or noisy topology?
2. Can wavelet-based calibration transfer to edge or graph classification tasks?
3. Does precomputing wavelets limit adaptability to dynamic graphs?
4. Could you briefly explain the intuition using wavelets to design a calibration algorithm in GNN?

**Details Of Ethics Concerns:**

No.

---

> ### Author Response · Authors · 2025-11-21
>
> # Weakness 1: Heterophilous graphs.
>
>
> We agree that WATS leverages the correlation between graph structure and calibration error. To assess how strong this dependency needs to be, we deliberately included heterophilous graphs in our benchmark and evaluated WATS there.
>
>
> We acknowledge that WATS leverages correlations between graph structure and calibration error. Empirically, this assumption is strongly supported by our results: on Cora-Full, a much larger and structurally more complex citation graph, WATS consistently achieves lower ECE than global TS and other graph-aware calibrators for both GCN and GAT backbones.
>
>
> Moreover, on heterophilous graphs such as Roman and Tolokers, WATS again delivers the best or tied-best ECE across all methods considered, indicating that structural information remains informative for calibration beyond the homophilous regime. Our ablations further show that wavelet-based multi-scale structural features outperform simpler structural descriptors (e.g., degree, centrality, clustering) on these datasets, reinforcing that the structure–calibration relationship exploited by WATS is robust across diverse graph types.
>
>
>
>
>
>
> # Weakness 2: Parameter tuning
>
>
> We note that the diffusion scale sss is indeed a hyperparameter, but its tuning cost is modest even on large graphs. In WATS, the computationally dominant step is computing the Chebyshev polynomials of the Laplacian, which costs O(k∣E∣)O(k|E|)O(k∣E∣) and is independent of s; varying the scale only changes the scalar coefficients in the linear combination of these precomputed terms, so different sss values can be evaluated very cheaply.
>
>
> In our sensitivity study we perform an exhaustive grid search over k∈{2,3,4} and s∈{0.4,0.8,1.2,1.6,2.0,2.5,3.0,4.0} on nine datasets, including large graphs such as Cora-Full and Reddit, and observe that WATS remains robust across a broad range of scales and consistently surpasses prior SOTA.
>
>
> We further prove the cost is reasonable by a table of computional time and cost.
>
>
> | Dataset | WATS Feature Extract Time | WATS Feature Extract Memory | WATS Calib Time | WATS Calib Memory | GETS Calib Time | GETS Calib Memory | GATS Calib Time | GATS Calib Memory | TS Calib Time | TS Calib Memory | CaGCN Calib Time | CaGCN Calib Memory |
> | :--- | ---: | ---: | ---: | ---: | ---: | ---: | ---: | ---: | ---: | ---: | ---: | ---: |
> | **Cora** | 0.0624 s | 207.31 MB | 1.0134 s | 207.92 MB | 22.2517 s | 97.39 MB | 2.1201 s | 95.00 MB | 0.3941 s | 94.65 MB | 4.1283 s | 94.65 MB |
> | **Computer** | 0.2374 s | 207.83 MB | 0.7716 s | 211.62 MB | 10.8242 s | 195.91 MB | 5.0126 s | 313.41 MB | 0.2408 s | 172.45 MB | 4.9372 s | 172.46 MB |
> | **Cora-Full** | 0.3134 s | 1955.27 MB | 1.0461 s | 1963.90 MB | 9.2106 s | 1401.74 MB | 5.2835 s | 1349.16 MB | 0.4754 s | 1344.47 MB | 4.7652 s | 1343.71 MB |
> | **Reddit** | 42.6636 s | 12456.00 MB | 1.3149 s | 4322.00 MB | 45.2529 s | 5938.96 MB | NAN | >17.54 GiB | 1.1054 s | 3855.37 MB | 20.8815 s | 3857.81 MB |
>
>
> Furthermore, we appreciate that the reviewer has also listed as a strength of our submission the detailed practical tuning guide we provide a range of k and s where surpass the current methods
>
> # Question 1: heterophily or noisy topology
>
> Our experiments indicate that WATS is robust to both heterophilous and noisy graph structure.
>
>
> Heterophily: We explicitly include the heterophilous benchmarks Roman and Tolokers in our evaluation, and on both datasets WATS consistently achieves the lowest ECE  compared with most of methods for both GCN and GAT backbones.
>
>
> Noisy topology: WATS also performs strongly on larger, structurally more irregular graphs such as Cora-Full and Reddit, where edges are naturally noisy and long-range connections are abundant. Here WATS again yields clear ECE improvements over global TS and prior graph-aware calibrators. We attribute this robustness to the use of wavelets, which aggregate structural information via diffusion and thus naturally smooth out local topological noise.

---

> > ### Author Response · Authors · 2025-11-21
> >
> > # Question 2:  Edge or graph classification tasks?
> > Thank you for pointing out an interesting direction of our work.
> > Our current work focuses on transductive node classification, but the underlying idea of wavelet-based structural calibration is not restricted to this setting. Conceptually, WATS always learns a task-specific temperature as a function of appropriate structural descriptors.
> >
> >
> > For node classification these descriptors are node-wise wavelet features; for edge classification one could analogously use wavelet features of the incident nodes; and for graph classification one could construct graph-level wavelet summaries (e.g., global pooling or histograms of node-wise wavelet responses) and learn a single temperature per graph, which may be similar to the temperature scaling but with graph specific features.
> >
> >
> > In all cases, the calibration module remains post-hoc and architecture-agnostic. We will add this generalization to edge and graph classification as an interesting avenue for future work.
> >
> >
> > # Question 3: Dynamic graphs
> >
> >
> > Precomputing wavelets does assume a static or slowly-changing graph. For dynamic graphs, there are two avenues:
> > Recompute wavelets periodically when topology changes significantly, which is still feasible if changes are not too frequent.
> >
> >
> > Use incremental / localized updates based on the sparse structure of the Laplacian (e.g., re-computing wavelets only for affected subgraphs), which we view as a promising future extension.
> >
> >
> > We will explicitly acknowledge this limitation and briefly sketch these directions in the discussion.
> >
> >
> >
> > # Question 4: Intuition of using wavelets
> >
> >
> > We thank the reviewer for this question. The intuition behind using wavelets is grounded in three key empirical observations regarding the limitations of current GNNs and simple structural statistics:
> >
> >
> > 1. Shallow Architectures Leave an Information Gap First, we observe that GNNs typically employ shallow architectures to avoid the "over-smoothing" phenomenon that degrades predictive performance in deep networks . However, this design choice comes at a cost: shallow GNNs are inherently unable to capture sufficient structural information beyond the immediate neighborhood . This leaves a "structural blind spot" in the model's confidence estimation.
> >
> >
> >
> >
> > 2. Low-Degree Nodes are Vulnerable Second, our analysis reveals a strong correlation between local connectivity and calibration error. Specifically, nodes with low degrees exhibit significantly larger Expected Calibration Error (ECE). This suggests that in regions where local structural information is sparse, the model's confidence becomes highly unreliable.
> >
> >
> > 3. Simple Signals (e.g., Degree) Are Insufficient Most importantly, we performed a cross-graph comparison and found that nodes with the exact same degree can exhibit vastly different ECEs across different graphs.
> >
> >
> > This proves that a simple, scalar signal like "degree" is insufficient to explain miscalibration. Two nodes may both have a degree of 2, but one might be in a homophilous cluster while the other is a bridge in a heterophilous graph.
> >
> >
> > Unlike simple degree statistics, graph wavelets act as a multi-scale "structural lens." They can distinguish between these complex topological contexts even when local degree statistics are identical, effectively recovering the "deeper structural information" missed by shallow GNNs to accurately diagnose miscalibration.
> >
> >
> > Thank you again for your insightful comments. Motivated by your suggestions, we have conducted extensive explanations for the WATS. If these efforts serve to alleviate your concerns, we would kindly ask for your support in adjusting the score accordingly.

---

> ### Author Response · Authors · 2025-11-23
>
> We thank you again for your time and constructive feedback. We wanted to follow up to see if our previous response have sufficiently addressed your concerns.
>
> We were encouraged to see that other reviewer found our clarifications and revisions sufficient to raise the score, and we sincerely hope that our updates also resolve the issues you raised. If there are any remaining questions or concerns, we would be more than happy to answer them in the discussion period.

---

> ### Author Response · Authors · 2025-11-28
>
> Dear Reviewer Wzpk,
> Thank you again for your constructive feedback on our paper.
> As the discussion period is coming to a close within one week, we were wondering if you have had a chance to review our response. We would effectively appreciate your feedback on whether our reduttal have resolved your concerns.
> Kind regards,
> Authors

---

### Official Review · Reviewer_jtMi · 2025-10-31

**Soundness:** 2
**Presentation:** 2
**Contribution:** 2
**Rating:** 4
**Confidence:** 3

**Summary:**

This paper introduces WATS, a post-hoc calibration framework for GNNs that learns node-wise learnable temperatures from graph wavelet-based structural features. By incorporating hierarchical structural signals, WATS improve miscalibration of GNNs compared to one-hop-based temperature scaling approaches.

**Strengths:**

- The application of graph wavelet transform in calibration domain is interesting.
- WATS substantially surpasses prior work in both small- and large-scale datasets.

**Weaknesses:**

- The discussion in Section 3.2 is a bit confusing. Even if model confidence exhibit high uncertainty in cases of disagreeing neighbors, a single node-level bias itself may not be problematic, since calibration error is inherently a population-level quantity, not measurable at a single-node level in practice. Could the authors provide additional explanation on this?
- According to Figure 1, deeper GCNs become more confident but less correct, which seems that increasing the receptive field rather worsens calibration. Furthermore, such phenomenon is likely attributable to over-smoothing rather than multi-scale connectivity or structural differences. This interpretation appears sometwhat misaligned with the design philosophy of WATS, which assumes that calibration errors arise when the calibration network fails to capture higher-order context. Could the authors clarify this?
- Minor) Found a typo in line 128.

**Questions:**

- Does similar trends of accuracy and confidence according to different numbers of layers in Figure 1 persist under heterophilous graphs?
- Could the authors show the wall-clock time analysis and memory consumption of the proposed method compared with baselines?

---

> ### Author Response · Authors · 2025-11-21
>
> # Weakness 1: Clarification on Section 3.2
> We apologize for the confusion caused by the discussion in Section 3.2.
>
>
> Our intention with the "per-node calibration bias" ($bias_i$) introduced in Eq. 3 was to serve as a simplified theoretical proxy to illustrate the mechanism by which 1-hop methods fail, rather than to propose a node-level metric to be minimized directly.
>
>
> We used this simplified scenario to demonstrate that in sparse or low-homophily regions, one-hop statistics often become "uninformative" (e.g., averaging to a constant regardless of the true label), which guarantees a high theoretical error for that specific node.
>
>
> The critical link to population-level calibration is that this failure is systematic:
>
>
> 1.  Because 1-hop statistics fail to capture the true structural context, this high theoretical bias applies not just to one node, but to entire sub-populations of nodes sharing these structural characteristics.
> 2.  When these systematically biased nodes are aggregated into bins, they inevitably result in a high ECE for the population.
>
>
> Therefore, this simplified node-level analysis was intended to provide the fairly natural motivation for why aggregating richer signals beyond immediate neighbors  is a necessary condition for reducing population-level calibration error.
>
>
> We will revise the discussion around Figure 1 to emphasize this separation and to clarify that WATS is designed precisely to avoid the pitfalls seen when the receptive field is increased purely via deeper GCNs.
>
>
> # Weakness 2: Why Wavelets rather than deeper GNN
>
>
> We thank the reviewer for this insightful comment. We agree that the miscalibration in deep GNNs (Figure 1) is largely driven by over-smoothing. However, this actually reinforces the necessity of our approach.
>
>
> 1. Over-smoothing is Feature-Based:
> As defined in the literature[1], over-smoothing is characterized by node features ($\mathbf{X}$) becoming indistinguishable, measured by the minimization of Dirichlet energy. This renders the GNN "blind" to node differences, leading to high-confidence errors.
>
>
> 2. WATS is Decoupled and Robust:
>
> Crucially, WATS does not rely on these homogenized features. Instead, it uses graph wavelets based on log-degree—a static, purely structural signal. This makes WATS immune to the feature over-smoothing that degrades the GNN.
>
>
> Therefore, WATS does not deepen the prediction GNN nor introduce additional message passing. The backbone GCN/GAT is kept fixed and its node features and logits remain unchanged. WATS only attaches a lightweight post-hoc calibration module that consumes graph wavelet features. Unlike simply stacking more GNN layers, WATS cannot make the task features less discriminative or aggravate over-smoothing; it instead increases structural awareness at the calibration stage, using multi-scale structural descriptors to assign node-wise temperatures and correct the depth-induced, graph-structured miscalibration observed in Fig. 1.
>
>
> [1] Rusch, T. K., Bronstein, M. M., & Mishra, S. (2023). A survey on oversmoothing in graph neural networks. arXiv preprint arXiv:2303.10993.
>
>
> # Weakness 3: Typo
> Thank you and we will correct the indicated typos.
>
>
> # Question 1: Trends on Heterophilous Graphs
>
>
> To answer this, we extended our depth analysis to the heterophilous Roman-empire dataset.
>
>
> Simply put, we observed a similar overall trend where model performance degrades with depth. Most notably, accuracy exhibits a significant decline.
>
>
> | Metric | 2-Layer | 4-Layer | 6-Layer | Trend Analysis |
> | :--- | :--- | :--- | :--- | :--- |
> | **Accuracy** | **37.5%** | 29.3% | 24.3% | **Significant Decline** |
> | **Confidence** | 32.9% | 32.7% | 28.0% | **Slight Decrease** |

---

> > ### Author Response · Authors · 2025-11-21
> >
> > # Question 2: Wall-clock time and memory
> > We thank the reviewer for this excellent suggestion. We agree that providing concrete performance metrics is essential to substantiate our claims of efficiency and scalability. We have conducted the requested end-to-end analysis, and the results strongly support our method's design.
> >
> >
> > We have compiled the wall-clock time and peak GPU memory usage on several datasets of varying sizes. The full results are presented below:
> >
> >
> > | Dataset | WATS Feature Extract Time | WATS Feature Extract Memory | WATS Calib Time | WATS Calib Memory | GETS Calib Time | GETS Calib Memory | GATS Calib Time | GATS Calib Memory | TS Calib Time | TS Calib Memory | CaGCN Calib Time | CaGCN Calib Memory |
> > | :--- | ---: | ---: | ---: | ---: | ---: | ---: | ---: | ---: | ---: | ---: | ---: | ---: |
> > | **Cora** | 0.0624 s | 207.31 MB | 1.0134 s | 207.92 MB | 22.2517 s | 97.39 MB | 2.1201 s | 95.00 MB | 0.3941 s | 94.65 MB | 4.1283 s | 94.65 MB |
> > | **Computer** | 0.2374 s | 207.83 MB | 0.7716 s | 211.62 MB | 10.8242 s | 195.91 MB | 5.0126 s | 313.41 MB | 0.2408 s | 172.45 MB | 4.9372 s | 172.46 MB |
> > | **Cora-Full** | 0.3134 s | 1955.27 MB | 1.0461 s | 1963.90 MB | 9.2106 s | 1401.74 MB | 5.2835 s | 1349.16 MB | 0.4754 s | 1344.47 MB | 4.7652 s | 1343.71 MB |
> > | **Reddit** | 42.6636 s | 12456.00 MB | 1.3149 s | 4322.00 MB | 45.2529 s | 5938.96 MB | NAN | >17.54 GiB | 1.1054 s | 3855.37 MB | 20.8815 s | 3857.81 MB |
> >
> >
> > Our implementation uses sparse operations for wavelets and a small MLP, so we expect WATS to be faster than heavy graph-specific calibrators in practice with reasonable computation cost.
> >
> > Thank you again for your insightful comments. Motivated by your suggestions, we have conducted extensive explanations for the WATS. If these efforts serve to alleviate your concerns, we would kindly ask for your support in adjusting the score accordingly.

---

> ### Author Response · Authors · 2025-11-23
>
> We thank you again for your time and constructive feedback. We wanted to follow up to see if our previous response have sufficiently addressed your concerns.
>
> We were encouraged to see that other reviewer found our clarifications and revisions sufficient to raise the score, and we sincerely hope that our updates also resolve the issues you raised. If there are any remaining questions or concerns, we would be more than happy to answer them in the discussion period.

---

> ### Author Response · Authors · 2025-11-28
>
> Dear Reviewer jtMi,
> Thank you again for your constructive feedback on our paper.
> As the discussion period is coming to a close within one week, we were wondering if you have had a chance to review our response. We would effectively appreciate your feedback on whether our reduttal have resolved your concerns.
> Kind regards,
> Authors

---

### Official Review · Reviewer_md66 · 2025-10-31

**Soundness:** 3
**Presentation:** 3
**Contribution:** 3
**Rating:** 6
**Confidence:** 4

**Summary:**

This paper focuses on the problem of calibrating model outputs for GNN architectures. Recent work shows that GNNs tend to be underconfident, which sets them apart from a lot of the NN literature and calls for bespoke strategies. A few strategies have already been suggested, that essentially try to predict the temperature parameter based on graph features. The central contribution of this paper is to replace this by a graph wavelet transform. The authors then show that their method performs well (where performance is measured using ECE) in several graphs, and outperforms competitors.

**Strengths:**

1) The paper's exposition is clear.
2) The results on experimental data look convincing.

**Weaknesses:**

As a disclaimer, I am not an expert on calibration (let alone on the calibration of GNNs), so here are a couple of points that I was hoping the authors could comment on.

1) Overall, the idea of using the graph structure to calibrate the uncertainty is fairly natural, but I wonder if the authors could try to phrase mathematically what they are trying to achieve.
For instance: I could imagine a setting where the GNN doesnt account perfectly for the signal on the graph, and as a consequence, the residuals are correlated.  For instance, let's simplify things by assuming that you're doing regression to predict $y_i = (\alpha X_i + \beta (\sum_{j \sim i} X_i/ d_i)  + \epsilon_i$  but that you're not using the graph structure, so $\hat{y}_i = \hat{\alpha} X_i$.

Consequently, using the wavelets that you are using is a much better idea than using predictors (like degree), as the wavelets will encode some deeper structural information (e.g where the errors are relative to one another) that will probably allow to calibrate more accurately the error, without having the overhead of expensive computations like GAT, etc.
Another scenario could be that you've explained the graph structure away using your GNN, but the errors $\epsilon_i$ are not iid (network effects --- e.g. friends talk to one another and exert mutual influence on each other). In this case, maybe a way of formulating the problem is that each node has a neighborhood effect:
$y_i = (\alpha X_i + \beta (\sum_{j \sim i} X_i/ d_i)  +  u_i + \epsilon_i$
where $u_i \sim N(0, \Sigma)$ where $\Sigma$ encodes dependencies between nodes. It is probably the case that the architecture you're suggesting is able to find dependencies that allow you to account for this random effect.
Is any of these close to what you were envisioning for the effect of the wavelets? Can you explain why the wavelet is a better idea?

2) The authors discuss the computational complexity of the method. It would have been insightful to compare the running time of each methods as well, on top of reporting the ECE.

3) It looks like the model doesnt crucially depend on K or s. What if, instead of the spectral wavelets, the authors used the Laplacian embedding of the graph --- that would prevent them from computing huge svd (just keep the top K) --- would we have similar results?

**Questions:**

(1) The characterization of conformal prediction as an "in-training approach [that]  uncertainty estimation within the model optimization process" seems off. CP acts as a wrapper, with no need to train any algorithm, so it doesnt interfer with the optimization process at all. It belongs to the post hoc methods.

(2) Could you explain this sentence: "GNNs tend to be systematically underconfident: their predicted confidence scores are consistently
lower than their true accuracy" --- what does "predicted confidence scores" mean, computed how?

(3) Tables 7-9 seem to show that the performance of the method is independent of the temperature parameter $s$ and $k$ -> any insight from the theory perspective?

4) For the ablation study, why not use the features? (Im not necessarily asking for more experiments, just curious to hear why it was not considered)?


Notes: Line 68: ".Differs" --- the full stop there seems to be an error.
Line 269: "The hyper-parameter k sets the maximum receptive-field size" -> k should probably be K?

---

> ### Author Response · Authors · 2025-11-21
>
> # Weakness 1: Mathematical Intuition
> We thank the reviewer for this thoughtful mathematical framing. Both scenarios you describe are closely aligned with our thinking, and the first scenario matches our design motivation most directly.
> Specifically, we view WATS as a mechanism to help the GNN utilize "deeper graph structural information" for calibration. Standard GNNs often fail to preserve complex topological signals, due to over-smoothing or limited receptive fields—effectively leaving a "structural residual" in the predictions. WATS leverages graph wavelets to explicitly recover this omitted information, providing a multi-scale geometric context that acts as a robust proxy for the signal the GNN missed.This concept of "deeper structure" directly enables us to model the network effects described in the second scenario with fine-grained sensitivity.
> By recovering this information, WATS can detect subtle variations in how errors propagate through the graph. For instance, by controlling the Chebyshev polynomial order $k$, we can explicitly regulate the range of these network effects.
>
>
> A small $k$ allows the model to focus on local, high-frequency effects (such as immediate neighbor disagreement), while a larger $k$ captures broader, meso-scale error propagation. This tunability allows WATS to adaptively sense the specific error covariance structure inherent to the dataset. Our ablation studies provide strong empirical evidence for the necessity of this approach.
> As shown in Table 3, graph wavelets consistently yield superior calibration performance, proving their effectiveness in diagnosing GNN miscalibration. Crucially, our experiments demonstrate that simply stacking coarse graph features (such as Degree, Centrality, and Clustering Coefficient) cannot effectively capture this "deeper graph structural information" These simple aggregates fail to represent the complex, multi-hop dependencies that wavelets naturally encode, confirming that the specific multi-scale spectral representation used in WATS is essential for accurate calibration.
> We will add a short explanation in the revision to make this intuition clear and to connect WATS more directly to this structural residual perspective.
>
>
> # Weakness 2: Computational complexity
> We thank the reviewer for this excellent suggestion. We agree that providing concrete performance metrics is essential to substantiate our claims of efficiency and scalability. We have conducted the requested end-to-end analysis, and the results strongly support our method's design.
>
>
> We have compiled the wall-clock time and peak GPU memory usage on several datasets of varying sizes. The full results are presented below:
>
>
> | Dataset | WATS Feature Extract Time | WATS Feature Extract Memory | WATS Calib Time | WATS Calib Memory | GETS Calib Time | GETS Calib Memory | GATS Calib Time | GATS Calib Memory | TS Calib Time | TS Calib Memory | CaGCN Calib Time | CaGCN Calib Memory |
> | :--- | ---: | ---: | ---: | ---: | ---: | ---: | ---: | ---: | ---: | ---: | ---: | ---: |
> | **Cora** | 0.0624 s | 207.31 MB | 1.0134 s | 207.92 MB | 22.2517 s | 97.39 MB | 2.1201 s | 95.00 MB | 0.3941 s | 94.65 MB | 4.1283 s | 94.65 MB |
> | **Computer** | 0.2374 s | 207.83 MB | 0.7716 s | 211.62 MB | 10.8242 s | 195.91 MB | 5.0126 s | 313.41 MB | 0.2408 s | 172.45 MB | 4.9372 s | 172.46 MB |
> | **Cora-Full** | 0.3134 s | 1955.27 MB | 1.0461 s | 1963.90 MB | 9.2106 s | 1401.74 MB | 5.2835 s | 1349.16 MB | 0.4754 s | 1344.47 MB | 4.7652 s | 1343.71 MB |
> | **Reddit** | 42.6636 s | 12456.00 MB | 1.3149 s | 4322.00 MB | 45.2529 s | 5938.96 MB | NAN | >17.54 GiB | 1.1054 s | 3855.37 MB | 20.8815 s | 3857.81 MB |
>
> Our implementation uses sparse operations for wavelets and a small MLP, so we expect WATS to be faster than heavy graph-specific calibrators in practice.

---

> ### Author Response · Authors · 2025-11-21
>
> # Weakness 3: Laplacian embeddings
> We thank the reviewer for this suggestion. A fuller discussion of K and s is provided in our response to Question 3. Regarding Laplacian embeddings, top K eigenvectors mainly capture global smooth structure, while heat kernel wavelets provide localized and scale dependent information. Since miscalibration often varies across local regions of the graph, these localized descriptors are more suitable for our calibration setting. We will clarify this distinction in the revision.
>
>
> Additionally, wavelets can be implemented without full eigendecomposition, using Chebyshev polynomial approximations, which avoids the heavy SVD often required for Laplacian embeddings on large graphs.
>
>
> We thus expect wavelets to be more robust on large and irregular graphs where global eigenvectors might be dominated by a few large components.
>
>
> We will mention this alternative in the discussion and include a small ablation comparing wavelet features vs. truncated Laplacian eigen-embeddings (top 10) to empirically support this point.
> | Dataset | lap_emb  (Mean ± Std Dev) | WATS (Mean ± Std Dev) |
> |---|---|---|
> | cora | 3.08 ± 1.21 | **1.82 ± 0.27** |
> | citeseer | 2.94 ± 0.52 | **2.11 ± 0.43** |
> | computers | 3.61 ±0.43 | **1.20 ± 0.19** |
> | pubmed | 2.58 ± 0.41 | **1.12 ± 0.09** |
> | photo | 1.75 ± 0.22 | **1.64 ± 0.31** |
> | cora-full | 5.72 ± 0.13 | **1.94 ± 0.11** |
> | reddit | 2.03 ± 0.10 | **0.90 ± 0.05** |
> | roman | 4.04 ± 0.31 | **3.42 ± 0.77** |
> | tolokers |$2.96 ± 1.44 | **2.45 ± 0.22** |
>
>
>
>
> # Question 1: Error in literature review
>
>
> Thank you for this correction. That was an error in our categorization. Conformal prediction is indeed a post-hoc wrapper, not an in-training method that modifies the optimization process. Our text incorrectly grouped it with the in-training approaches. We will move it to the post-hoc methods section in the revision.
>
>
> # Question 2: Predicted confidence scores
>
>
> By “predicted confidence scores,” we mean the softmax probability assigned to the predicted class for each node. “Underconfidence” means that, when we group nodes by their predicted confidence, the empirical accuracy in each group tends to be higher than that confidence (i.e., the reliability curve lies above the diagonal). We will rewrite this sentence to avoid ambiguity and reference the specific plots showing this effect.
>
>
> # Question 3: Predicted confidence scores
>
>
> We thank the reviewer for this insightful observation regarding the hyperparameter sensitivity in Figure 3. We respectfully clarify that the apparent "independence" of performance from parameters $s$ and $k$ is not a universal property of the method, but rather a reflection of the structural simplicity of specific datasets.
> The results actually reveal a critical distinction: while WATS exhibits desirable robustness on simple, homophilous graphs, its tunability is theoretically and empirically essential for handling complex, heterophilous structures.
>
>
> It is true that on datasets like Cora and Computers, the performance curves become relatively stable when$s > 1.2$. This "independence" is physically expected: these graphs are highly homophilous, meaning the structural signal governing uncertainty is predominantly low-frequency (smooth). As long as the diffusion scale $s$ is sufficiently large to cover the local neighborhood, the wavelet transform consistently captures this coarse-grained signal. This should be interpreted as stability, demonstrating that WATS is not brittle in standard scenarios.
>
>
> In contrast, the "independence" hypothesis completely breaks down on lower-homophily graphs, validating the necessity of our tunable design:
> Sensitivity to Over-smoothing (Cora-Full): On the Cora-Full dataset, we observe a distinct upward trend where ECE degrades significantly—from approximately 1.95 at $s=0.4$ to over 2.60 at $s=4.0$. This confirms that "over-smoothing" the calibration signal via a large $s$ is detrimental here. A smaller $s$ (restricted diffusion) is strictly necessary to preserve the high-frequency structural details (e.g., local heterophily) that drive miscalibration in this dataset.
>
>
> The "Sweet Spot" for Band-Pass Filtering (Roman): The heterophilous Roman dataset provides the strongest counter-evidence. The performance is highly sensitive to both parameters. The $k=4$ model (green line) exhibits a sharp "V-shape," achieving a minimum ECE of 3.43 at exactly $s=2.5$, whereas other settings (such as $k=2$) fail to drop below 3.8. This indicates that calibration on heterophilous graphs requires a specific spectral band-pass filter: we need a larger receptive field ($k=4$) to reach relevant context, combined with a specific diffusion scale ($s=2.5$) to filter out long-range noise.
>
>
> We will add a more detailed discussion in hyper-parameter section

---

> ### Author Response · Authors · 2025-11-21
>
> # Question 4: Node features
> We focused the ablation on structural signals (degree, centrality, etc.) because WATS is intended to be:
>
>
> Our main motivation is to investigate if structural signals can improve GNN calibration
>
>
> Label-free and feature-agnostic: applicable when features are high-dimensional, partially missing, or sensitive; and
>
>
> Cleanly separated from the backbone’s feature processing.
>
>
> However, we agree that an ablation using features would be informative. We will note this in the paper and, if space allows, include a variant where wavelet features are concatenated with low-dimensional feature summaries to show how much additional gain this provides.
>
>
> | Dataset | WATS (Mean ± Std Dev) | feature (Mean ± Std Dev) |
> |---|---|---|
> | cora | **1.82 ± 0.27** | 3.36 ± 0.51 |
> | citeseer | **2.11 ± 0.43** | 5.20 ± 1.22 |
> | computers | **1.20 ± 0.19** | 3.42 ± 0.52 |
> | pubmed | **1.12 ± 0.09** | 2.30 ± 0.36 |
> | photo | **1.64 ± 0.31** | 1.68 v 0.23 |
> | cora-full | **1.94 ± 0.11** | 5.56 ± 0.21 |
> | reddit | **0.90 ± 0.05** | 1.99 ± 0.21 |
> | roman | **3.42  ± 0.77** | 3.94 ± 0.55 |
> | tolokers | 2.45 ± 0.22 | **2.23 ± 0.98** |
>
>
> # Note typo
> We will correct the indicated typos and notation inconsistencies.
>
>
> Thank you again for your insightful comments. Motivated by your suggestions, we have conducted extensive new experiments to verify our method. If these efforts serve to alleviate your concerns, we would kindly ask for your support in adjusting the score accordingly.

---

> > ### Comment · Reviewer_md66 · 2025-11-23
> > **Re:**
> >
> > The authors have put in a tremendous amount of work to further test and benchmark their method. Their idea sounds interesting and seems to be working well in practice. While I would have appreciated a little bit more of a mathematical formulation to the objective, I like the paper and think it contributes to the literature enough to warrant acceptance. I am raising my score.

---

> > > ### Author Response · Authors · 2025-11-23
> > > **Thank you!**
> > >
> > > We sincerely thank the you for raising the score and for your appreciation of our work! We are also grateful for your valuable suggestions, which we will incorporate into the revised manuscript.

---

### Official Review · Reviewer_Cr2Z · 2025-11-01

**Soundness:** 3
**Presentation:** 3
**Contribution:** 2
**Rating:** 4
**Confidence:** 4

**Summary:**

The authors aim to address the calibration issue in node classification. Existing methods only consider local topology when performing calibration, which limits their effectiveness. To overcome this issue, the author proposes using graph wavelets, which can capture information from more distant nodes, to determine the temperature for calibration. Experimental results demonstrate that the proposed method outperforms existing approaches on both GCN and GAT models.

**Strengths:**

- Proposes a method that can consider beyond one-hop neighborhood information
- Demonstrates the effectiveness of the proposed method across datasets with diverse characteristics
- The approach is lightweight, as it only requires running a 2-layer MLP on the validation set

**Weaknesses:**

- The method novelty of the proposed approach appears limited. Graph wavelets are already well-established, and in this work, they are applied to the calibration setting with only minor engineering and no specialized adaptation, which diminishes the originality of the method.
- The method involves training a MLP on the validation set, where the current split allocates 10% to validation and 20% to training. This is a relatively large validation portion, and in practical scenarios, such a large validation set may not be feasible. Consequently, the performance might degrade as the validation size decreases.
- Experiments are conducted only on relatively old architectures such as GCN and GAT, with no evaluation on recent GNN architectures, particularly those that include skip connections. Thus, it remains unclear whether the proposed method generalizes well to modern GNNs.
- Although the paper emphasizes the advantage of capturing beyond one-hop information, the evaluation is limited to 2-layer GCN and GAT, making it uncertain whether the method performs equally well on deeper architectures.

**Questions:**

Please see the Weakness.

---

> ### Author Response · Authors · 2025-11-21
>
> # Weakness 1: Novelty
> We thank the reviewer for their feedback and wish to clarify our method's novelty by focusing on the three points you outlined. We believe these points directly address the concern about "minor engineering" and demonstrate a clear, specialized contribution.
>
>
> Existing methods (e.g., GATS, CaGCN) are limited, "primarily depend on coarse one-hop statistics". WATS is the first to posit that GNN miscalibration is a multi-scale problem driven by "fine-grained structural heterogeneity" , requiring a solution that goes "beyond local neighbor statistics".
>
>
> Our choice of log-degree as the base signal is a key adaptation, not a default. Table 2 proves this is a better base signal compared to using raw 'Degree' or an 'Identity Matrix'.
>
>
> Through ablation studies, we also empirically validate the performance advantage of WATS compared to standard topological descriptors. These results highlight the necessity of encoding deeper structural information for effective GNN calibration.
>
>
> We will add more detail to  distinguish WATS from prior graph calibration approaches in the revised version.
>
>
> # Weakness 2: Train test split
> We appreciate this practical concern. Our choice mirrors common calibration practice, including in prior GNN calibration work(Hsu et al., 2022; Tang et al., 2024; Zhuang et al., 2024 ), but we agree that robustness to smaller validation sets is important.
>
>
> Conceptually, WATS learns a low-capacity calibrator (2-layer MLP) on relatively low-dimensional wavelet descriptors, which should be less prone to overfitting than methods that learn larger models over logits or neighbor features.
>
>
> We also add and report how ECE and accuracy change for WATS and baselines when 15% for training and 5% for validation. This will directly answer how performance scales with validation size.
>
>
> As shown in the updated table, WATS demonstrates superior data efficiency. It retains the State-of-the-Art (SOTA) performance on 7 out of 9 datasets, consistently outperforming other baselines.
> | Dataset   | Uncali          | TS             | ETS            | CaGCN          | GATS            | GETS            | WATS            |
> | :---      | :---            | :---           | :---           | :---           | :---            | :---            | :---            |
> | cora      | 23.67 ± 1.28    | 2.47 ± 0.41    | 2.75 ± 0.57    | 2.70 ± 0.58    | **1.96 ± 0.24** | 3.01 ± 0.50     | 2.67 ± 0.44     |
> | citeseer  | 25.59 ± 3.35    | 5.92 ± 1.20    | 5.93 ± 1.19    | 6.30 ± 1.18    | 5.08 ± 0.90     | **3.84 ± 1.44** | 5.20 ± 1.64     |
> | computers | 6.92 ± 1.16     | 4.01 ± 0.8     | 3.98 ± 0.95    | 3.13 ± 1.11    | 2.73 ± 0.70     | 8.08 ± 7.33     | **1.45 ± 0.58** |
> | pubmed    | 14.27 ± 1.20    | 2.49 ± 0.33    | 2.57 ± 0.36    | 1.45 ± 0.56    | 2.10 ± 0.35     | 2.34 ± 0.47     | **1.13 ± 0.12** |
> | photo     | 3.67 ± 0.84     | 1.66 ± 0.54    | 1.43 ± 0.21    | 1.53 ± 0.20    | 1.54 ± 0.29     | 1.65 ± 0.88     | **1.39 ± 0.21** |
> | cora-full | 27.41 ± 0.22    | 5.07 ± 0.13    | 5.08 ± 0.13    | 4.47 ± 0.59    | 5.13 ± 0.13     | 3.09 ± 0.80     | **2.58 ± 0.32** |
> | reddit    | 6.66 ± 0.13     | 1.75 ± 0.06    | 1.75 ± 0.06    | 1.38 ± 0.07    | Nan             | 2.62 ± 0.35     | **1.02 ± 0.20** |
> | roman     | 9.49 ± 0.27     | 4.30 ± 0.27    | 4.30 ± 0.27    | 5.46 ± 0.58    | 4.45 ± 0.31     | 5.21 ± 0.93     | **3.93 ± 0.92** |
> | tolokers  | 5.76 ± 2.05     | 3.62 ± 1.08    | 3.63 ± 1.08    | 2.39 ± 0.68    | 3.01 ± 0.74     | 3.85 ± 0.97     | **2.27 ± 0.38** |
>
>
> reference
>
>
> [1]Hsu, H. H. H., Shen, Y., Tomani, C., & Cremers, D. (2022). What makes graph neural networks miscalibrated?. Advances in Neural Information Processing Systems, 35, 13775-13786.
>
>
> [2]Zhuang, D., Jiang, C., Zheng, Y., Wang, S., & Zhao, J. (2024). GETS: Ensemble Temperature Scaling for Calibration in Graph Neural Networks. arXiv preprint arXiv:2410.09570.
>
>
> [3]Tang, B., Wu, Z., Wu, X., Huang, Q., Chen, J., Lei, S., & Meng, H. (2024, March). Simcalib: Graph neural network calibration based on similarity between nodes. In Proceedings of the AAAI Conference on Artificial Intelligence (Vol. 38, No. 14, pp. 15267-15275).

---

> ### Author Response · Authors · 2025-11-21
>
> # Weakness 3: Skip connections GNN model
> We fully agree that showing results on modern architectures would strengthen the paper.
> We added experiments on GNN(GCNII) with Skip connection .
> Emphasize that applying WATS to a new backbone requires no changes: we simply take its logits as input and reuse the same wavelet-based calibrator.
> | Dataset | uncali | TS | ETS | CaGCN | GATS | GETS | WATS |
> |---|---|---|---|---|---|---|---|
> | cora | 17.35 ± 3.28 | 3.38 ± 0.92 | 3.35 ± 0.93 | 4.35 ± 2.35 | 3.43 ± 1.07 | 6.76 ± 4.94 | **3.23 ± 1.01** |
> | citeseer | 13.32 ± 10.99 | 7.39 ± 4.38 | 7.43 ± 4.48 | 8.65 ± 1.77 | 8.66 ± 2.62 | **6.68 ± 3.42** | 7.27 ± 3.49 |
> | computers | 10.30 ± 1.37 | 6.78 ± 0.67 | 6.91 ± 0.87 | 5.69 ± 0.47 | 5.62 ± 0.56 | 2.89 ± 1.26 | **3.89 ± 0.68** |
> | pubmed | 12.94 ± 1.18 | 3.21 ± 0.91 | 3.65 ± 0.91 | **2.02 ± 1.67** | 2.42 ± 0.93 | 2.23 ± 0.31 | **2.10 ± 0.34** |
> | photo | 9.66 ± 1.27 | 1.65 ± 0.54 | 1.88 ± 0.66 | 1.50 ± 0.36 | 1.65 ± 0.56 | 3.15 ± 1.28 | **1.33 ± 0.48** |
> | cora-full | 15.68 ± 2.85 | 3.51 ± 0.62 | 3.50 ± 0.61 | 3.28 ± 0.97 | 3.50 ± 0.59 | 3.01 ± 0.88 | **2.92 ± 0.98** |
> | reddit | 17.73 ± 1.10 | 1.41 ± 0.36 | 1.44 ± 0.34 | 1.20 ± 0.28 | Nan | 2.99 ± 0.53 | **0.88 ± 0.28** |
> | roman | 21.00 ± 0.42 | 3.61 ± 0.65 | 3.61 ± 0.65 | 4.62 ± 0.96 | 4.38 ± 0.84 | 4.34 ± 1.18 | **2.92 ± 1.27** |
> | tolokers | 6.40 ± 0.62 | 4.39 ± 0.61 | 4.02 ± 0.33 | 4.38 ± 0.61 | 4.02 ± 0.33 | 4.41 ± 0.48 | **3.34 ± 0.20** | | $4.38 \pm 0.61$ | $4.02 \pm 0.33$ | $4.41 \pm 0.48$ | **$3.34 \pm 0.20$** |
>
>
> This will demonstrate that the gains are not tied to a specific backbone family. Even when the underlying GNN is designed with skip connections to handle depth and structural information better, WATS successfully extracts complementary topological signals to further refine confidence.
>
>
> # Weakness 4 Deeper Architectures
>
>
> We thank you for this question regarding the impact of GNN depth on calibration. Our main experiments used standard 2-layer GNNs, consistent with common practice for those benchmarks. To address your question, we conducted new experiments on deeper models.
>
>
> The results on the Computers dataset for 4-layer and 6-layer GNNs show that WATS remains the most effective calibration method, consistently achieving the lowest Expected Calibration Error (ECE).
>
>
> | GNN Depth | Uncalibrated | TS | GATS | CaGCN | GETS | WATS |
> | :--- | :--- | :--- | :--- | :--- | :--- | :--- |
> | **4 Layers** | 5.43 ± 1.35 | 4.91 ± 0.38 | 3.23 ± 1.39 | 2.91 ± 2.37 | 3.46 ± 1.16 | **2.81 ± 1.82** |
> | **6 Layers** | 5.33 ± 1.19 | 5.30 ± 1.10 | 3.51 ± 2.29 | 2.75 ± 1.18 | 5.64 ± 3.43 | **2.66 ± 0.72** |
>
>
> Thank you again for your insightful comments. Motivated by your suggestions, we have conducted extensive new experiments to verify our method. If these efforts serve to alleviate your concerns, we would kindly ask for your support in adjusting the score accordingly.

---

> ### Author Response · Authors · 2025-11-23
>
> We thank you again for your time and constructive feedback. We wanted to follow up to see if our previous response have sufficiently addressed your concerns.
>
> We were encouraged to see that other reviewer found our clarifications and revisions sufficient to raise the score, and we sincerely hope that our updates also resolve the issues you raised. If there are any remaining questions or concerns, we would be more than happy to answer them in the discussion period.

---

> ### Author Response · Authors · 2025-11-28
>
> Dear Reviewer Cr2Z,
> Thank you again for your constructive feedback on our paper.
> As the discussion period is coming to a close within one week, we were wondering if you have had a chance to review our response. We would effectively appreciate your feedback on whether our reduttal have resolved your concerns.
> Kind regards,
> Authors

---

### Author Response · Authors · 2025-12-02
**Rebuttal Summary**

Dear Program Chairs, Area Chairs, Senior Area Chairs, and Reviewers,

We sincerely thank the reviewers for their constructive feedback. We have addressed all the weaknesses and questions by conducting extensive new experiments (including tests on deeper GNNs, GCNII, smaller validation splits, and various ablation studies) and providing detailed clarifications, resulting in a **score improvement from (6, 6, 4, 4) to (8, 6, 4, 4)** before the OpenReview bug. We believe these efforts successfully resolve the primary concerns.

### **Rebuttal Time Line**
To prove that our rebuttal complies with standard academic disclosure and ethics, we provide the following timeline (AoE):

- Nov 11: Rebuttal period began.
- Nov 20: Submitted detailed rebuttal with experiments and explanations.
- Nov 23: Reviewer md66 **raised score from 6 to 8**.
- Nov 28: Officially disclosed the data leakage.
- Dec 3: Revised version uploaded.

### **Reviewer Cr2Z**
* **Novelty:** Clarified that WATS targets "multi-scale heterogeneity" and demonstrated that using log-degree is a key adaptation, not a default choice.
* **Robustness (Data Split):** Added experiments with a 15% training / 5% validation split, showing WATS retains SOTA performance on 7 out of 9 datasets.
* **Modern Architectures:** Verified performance on GCNII (with skip connections), proving WATS extracts complementary structural signals.
* **Deep GNNs:** Conducted experiments on 4-layer and 6-layer GNNs, where WATS consistently achieved the lowest ECE.

### **Reviewer md66 (Score raised from 6 to 8)**
* **Mathematical Intuition:** Explained WATS as a mechanism to recover the "structural residual"—deeper graph structural information lost by GNNs due to over-smoothing or limited receptive fields.
* **Efficiency:** Provided a wall-clock time and memory usage table, proving WATS is computationally efficient due to sparse operations.
* **Laplacian Embeddings:** Differentiated wavelets (localized/scale-dependent) from Laplacian embeddings (global), supported by an ablation study showing WATS performs significantly better.
* **Hyperparameter Sensitivity:** Clarified that parameter tuning is critical for heterophilous graphs to prevent over-smoothing, refuting the idea that performance is independent of parameters.
* **Literature Correction :** Acknowledged the error in categorizing Conformal Prediction and moved it to the post-hoc methods section.
* **Definitions :** Clarified "Predicted confidence scores" as softmax probabilities and defined "Underconfidence" regarding reliability curves.
* **Node Features Ablation :** Added an ablation study comparing WATS to using node features, showing WATS (structure-only) remains robust and often superior.

### **Reviewer jtMi**
* **Theoretical Clarification:** Clarified that the "per-node calibration bias" in Section 3.2 was a simplified theoretical proxy to explain why 1-hop methods fail, not a direct optimization metric.
* **Wavelets vs. Deep GNNs:** Argued that deepening GNNs causes feature over-smoothing, whereas WATS relies on static log-degree structure (decoupled from features) to remain robust and correct depth-induced errors.
* **Heterophilous Trends:** Extended depth analysis to the Roman-empire dataset, showing that accuracy declines significantly with depth in heterophilous settings.
* **Efficiency Metrics:** Provided the same computational performance table as requested by Reviewer 2.

### **Reviewer Wzpk**
* **Heterophilous Graphs:** Confirmed WATS is robust on heterophilous datasets (Roman, Tolokers), outperforming baselines.
* **Tuning Cost:** Explained that parameter tuning is cheap because the expensive Chebyshev polynomial computation is independent of the scale parameter $s$.
* **Dynamic Graphs:** Acknowledged the static graph limitation and proposed periodic re-computation or incremental updates as future work.
* **Task Generalization:** Outlined how the framework can be extended to edge classification (via incident nodes) and graph classification (via global summaries).
* **Outlined the core motivation:** Outlined the core motivation: Shallow GNNs leave a 'structural blind spot'; low-degree nodes are especially vulnerable to high ECE; and simple degree statistics are insufficient, which wavelets successfully address.

The discussion substantially strengthens the paper's theoretical and empirical foundation. We are confident in the effectiveness and efficiency of WATS and sincerely request that you consider acceptance of our work.

Kind regards,

Authors

---

### Meta-Review · Area_Chair_1Qkm · 2025-12-30

**Summary:**

This paper introduces Wavelet-Aware Temperature Scaling, a post-hoc calibration framework for node classification that leverages tunable heat-kernel graph wavelet features to assign node-specific temperatures, improving confidence estimation without model retraining or access to neighboring predictions.

**Strengths:**
(1) The proposed method is lightweight and easy to deploy.

(2) Its effectiveness is validated on a diverse set of datasets with comprehensive experimental evaluation.

(3) The application of graph wavelet transforms to the calibration setting is interesting and well-motivated.

**Weaknesses:**
(1) The level of methodological novelty appears somewhat limited, as graph wavelet transforms have been widely studied.

(2) The experiments rely on relatively small, shallow, and older backbone models, and lack comparisons with more recent baselines.

(3) The approach depends on the correlation between graph structure and calibration error.

Although the reliance on structural–calibration error correlation suggests that there may still be room for improvement on heterophilous graphs, the authors’ rebuttal convincingly addressed the other concerns through additional experiments and theoretical justification. Overall, the work is sufficiently solid and well-supported, and I therefore recommend acceptance.

**Reviewer Concerns:**

Reviewer Cr2Z
1. Novelty: Clarified that WATS targets "multi-scale heterogeneity" and demonstrated that using log-degree is a key adaptation, not a default choice. I believe this concern has been addressed to some extent.
2. Robustness (Data Split): Added experiments with a 15% training / 5% validation split, showing WATS retains SOTA performance on 7 out of 9 datasets. This concern has been adequately addressed.
3. Modern Architectures: Verified performance on GCNII (with skip connections), proving WATS extracts complementary structural signals. I believe this concern has been addressed to some extent.
4. Deep GNNs: Conducted experiments on 4-layer and 6-layer GNNs, where WATS consistently achieved the lowest ECE.  I believe this concern has been addressed to some extent.


Reviewer md66
1. Mathematical Intuition: Explained WATS as a mechanism to recover the "structural residual"—deeper graph structural information lost by GNNs due to over-smoothing or limited receptive fields. I believe this concern has been addressed to some extent.
2. Efficiency: Provided a wall-clock time and memory usage table, proving WATS is computationally efficient due to sparse operations. This concern has been adequately addressed.
3. Laplacian Embeddings: Differentiated wavelets (localized/scale-dependent) from Laplacian embeddings (global), supported by an ablation study showing WATS performs significantly better. This concern has been adequately addressed.
4. Hyperparameter Sensitivity: Clarified that parameter tuning is critical for heterophilous graphs to prevent over-smoothing, refuting the idea that performance is independent of parameters. This concern has been adequately addressed.
5. Literature Correction : Acknowledged the error in categorizing Conformal Prediction and moved it to the post-hoc methods section. This concern has been adequately addressed.
6. Definitions : Clarified "Predicted confidence scores" as softmax probabilities and defined "Underconfidence" regarding reliability curves. This concern has been adequately addressed.
7. Node Features Ablation : Added an ablation study comparing WATS to using node features, showing WATS (structure-only) remains robust and often superior. This concern has been adequately addressed.


Reviewer jtMi
1. Theoretical Clarification: Clarified that the "per-node calibration bias" in Section 3.2 was a simplified theoretical proxy to explain why 1-hop methods fail, not a direct optimization metric. I believe this concern has been addressed to some extent.
2. Wavelets vs. Deep GNNs: Argued that deepening GNNs causes feature over-smoothing, whereas WATS relies on static log-degree structure (decoupled from features) to remain robust and correct depth-induced errors. This concern has been adequately addressed.
3. Heterophilous Trends: Extended depth analysis to the Roman-empire dataset, showing that accuracy declines significantly with depth in heterophilous settings. I believe this concern has been addressed to some extent.
4. Efficiency Metrics: Provided the same computational performance table as requested by Reviewer 2. This concern has been adequately addressed.


Reviewer Wzpk
1. Heterophilous Graphs: Confirmed WATS is robust on heterophilous datasets (Roman, Tolokers), outperforming baselines. I believe this concern has been addressed to some extent.
2. Tuning Cost: Explained that parameter tuning is cheap because the expensive Chebyshev polynomial computation is independent of the scale parameter. This concern has been adequately addressed.
3. Dynamic Graphs: Acknowledged the static graph limitation and proposed periodic re-computation or incremental updates as future work. This concern has not been addressed.
4. Task Generalization: Outlined how the framework can be extended to edge classification (via incident nodes) and graph classification (via global summaries). I believe this concern has been addressed to some extent.
5. Outlined the core motivation: Outlined the core motivation: Shallow GNNs leave a 'structural blind spot'; low-degree nodes are especially vulnerable to high ECE; and simple degree statistics are insufficient, which wavelets successfully address. This concern has been adequately addressed.

**Reviewer Scores:**

Reviewer Cr2Z - I believe this reviewer is unlikely to raise his/her score, as he/she questions the novelty of the paper.

Reviewer md66 - I believe this reviewer will definitely raise his/her score (from 6 to 8), as he/she has already done so.

Reviewer jtMi - I believe this reviewer is very likely to raise their score.

Reviewer Wzpk - I believe this reviewer is very likely to raise their score.

---

### Decision · Program_Chairs · 2026-01-26

Accept (Poster)